# Distinct circadian mechanisms govern cardiac rhythms and susceptibility to arrhythmia

Edward A. Hayter [1], Sophie M. T. Wehrens [2], Hans P. A. Van Dongen [3,4], Alessandra Stangherlin [5], Shobhan Gaddameedhi[6], Elena Crooks[3,8], Nichola J. Barron[1], Luigi A. Venetucci[7], John S. O'Neill [5], Timothy M. Brown[1], Debra J. Skene[2], Andrew W. Trafford [1,7] & David A. Bechtold [1✉]

Electrical activity in the heart exhibits 24-hour rhythmicity, and potentially fatal arrhythmias are more likely to occur at specific times of day. Here, we demonstrate that circadian clocks within the brain and heart set daily rhythms in sinoatrial (SA) and atrioventricular (AV) node activity, and impose a time-of–day dependent susceptibility to ventricular arrhythmia. Critically, the balance of circadian inputs from the autonomic nervous system and cardiomyocyte clock to the SA and AV nodes differ, and this renders the cardiac conduction system sensitive to decoupling during abrupt shifts in behavioural routine and sleep-wake timing. Our findings reveal a functional segregation of circadian control across the heart's conduction system and inherent susceptibility to arrhythmia.

[1] Centre for Biological Timing, Faculty of Biology, Medicine and Health, University of Manchester, Manchester, UK. [2] Faculty of Health and Medical Sciences, University of Surrey, Guildford, UK. [3] Sleep and Performance Research Center, Washington State University, Spokane, WA, USA. [4] Elson S. Floyd College of Medicine, Washington State University, Spokane, WA, USA. [5] MRC Laboratory of Molecular Biology, Cambridge, UK. [6] Department of Biological Sciences, Center for Human Health and the Environment, North Carolina State University, Raleigh, NC, USA. [7] Unit of Clinical Physiology, Manchester Academic Health Science Centre, Faculty of Biology, Medicine and Health, University of Manchester, Manchester, UK. [8] Present address: Department of Physical Therapy, Eastern Washington University, Spokane, WA, USA. ✉email: David.bechtold@manchester.ac.uk

Pronounced time-of-day variation exists in most aspects of our cardiovascular physiology. This includes cardiac electro-physiology, where daily rhythms in heart rate (HR) and electrocardiogram (ECG) parameters can be readily observed in humans and laboratory animals[1]. Importantly, these temporal dynamics are not simply a consequence of the sleep–wake cycle or activity state, but are also heavily influenced by the body's internal circadian clock[2–4]. The circadian system is orchestrated by a master clock, located in the suprachiasmatic nuclei (SCN) of the hypothalamus, which directs gross rhythms in our physiology (e.g., daily sleep/wake, hormone, and temperature cycles) and keeps us aligned with the external light/dark environment. Circadian clocks also operate in most cells of the body. Therefore, daily fluctuation in tissue function reflects not only our behavioral state but also the activity of both central and local circadian clock function. As a prime example, the SCN contribute to rhythms in HR and heart rate variability (HRV) across the day via the autonomic nervous system[5,6], yet many aspects of cardiac function including contractility, metabolism, and even recovery from ischemic insult are strongly influenced by the local cardiomyocyte clock[3,4,7,8].

The time-of-day prevalence in many life-threatening cardiac rhythm disorders (e.g., extreme bradycardia, ventricular fibrillation, tachycardia, and sudden cardiac death) is well-documented[1]. Genetic disruption of clock function selectively in cardiomyocytes in mice can slow HR[7], lengthen QT interval[4], and alters ventricular expression of a number of ion channels responsible for electrical conduction in the heart (e.g., *Scn5a, Kcnh2*)[4,9,10]. Circadian misalignment can occur when our daily behavioral routine and/or external environment does not match our internal clock timing. We have recently shown that circadian misalignment driven by altered environmental light/dark cycles or imposed behavioral routine leads to altered cardiac electrophysiology in mice[11] and humans[12]. A role for the cardiomyocyte clock in cardiac arrhythmogenesis has also been previously implied, but never directly demonstrated[10]. Despite these findings, the source and mechanism of circadian control over the cardiac conduction system, and especially at the level of sinoatrial (SA) and atrioventricular (AV) function remain unclear. This is an important issue as conditions that disrupt our circadian system, such as shift work, are associated with increased incidence of cardiovascular events including conduction disorders[3,13–15].

Long-term continuous ECG recording and analyses provide a powerful and non-invasive method for determining cardiac conduction dynamics across time and in response to environmental or behavioral manipulations. Changes in HR and HRV have been associated with increased cardiovascular risk in numerous clinical conditions and in healthy individuals[16,17]. Beyond HR and HRV, ECG waveforms reveal underlying activity at the SA node (RR), atrial depolarization ($P_{wave}$), conduction delay at the AV node ($PR_{segment}$), ventricular depolarization (QRS), and ventricular repolarization time (QT) (Supplementary Fig. 1). Each of these parameters has important diagnostic value, and alterations therein are associated with an increased risk of potentially fatal conduction disturbance such as heart block (prolonged PR interval[18]) and sudden cardiac death (long QT[19]).

In this work, we investigate temporal aspects of cardiac electrophysiology in mice and humans, under normal conditions and in response to imposed shifts in behavioral routine (i.e., timing of sleep, activity, feeding, and ambient light exposure). These studies reveal distinct circadian and behavioral influence over aspects of cardiac function, as well as a pronounced impact of time-of-day and circadian clock function on susceptibility to cardiac arrhythmias.

## Results

### Mistimed sleep and behavioral cycles disrupt rhythms in SA and AV node activity in humans.
We first defined the influence of abrupt changes in the daily behavioral routine on cardiac electrophysiology using ECG recordings collected during our previous human sleep studies[20,21]. First, continuous ECG recordings were obtained from 25 healthy men (11 experienced shift workers, 14 age-matched controls) over a 4-day in-laboratory study[20]. Following 24 h of baseline recording, subjects underwent one night of total sleep deprivation (TSD), with a 4-h recovery nap the following afternoon (Fig. 1A). All participants remained in a semi-recumbent position during sleep episodes and equivalent waking periods on baseline and TSD days. HR and ECG characteristics were extracted and analyzed across the entire study. Robust diurnal rhythms in HR and many ECG characteristics (HRV, RR, QT, $PR_{seg}$ intervals) were observed in both control (Fig. 1B–F and Supplementary Fig. 2) and shift workers (Supplementary Fig. 3) under baseline conditions (black traces). ECG characteristics did not differ significantly between shift-work and control groups in terms of either 24 h mean or rhythmic profile at baseline (Supplementary Table 1).

In control (non-shift-work) individuals, TSD and subsequent daytime recovery nap led to an expected and significant response in HR and RR interval during TSD and recovery nap, compared to equivalent times on the baseline day (Fig. 1B, D and Supplementary Fig. 2). QT interval is strongly influenced by HR, and as such mirrored the changes in RR (Fig. 1E and Supplementary Fig. 2). By contrast, $PR_{seg}$ duration (reflecting conduction through the AV node) was insensitive to the change in behavioral state, and followed a consistent 24 h profile across baseline and TSD days (Fig. 1F and Supplementary Fig. 2). While we have previously reported a small difference in RR variability (SDNN) following sleep deprivation[20], no differences in HRV were found between baseline and TSD days using a geometric measure[22] which is robust against changes in absolute HR (Fig. 1C). This suggests that time-of-day variation in parasympathetic tone (thought to be largely responsible for the circadian rhythm in HRV[6]) was not disturbed by the change in arousal state. Thus, two critical aspects of cardiac conduction, SA node pacemaking, and AV nodal delay (reflected by RR and $PR_{seg}$ duration, respectively) exhibit clear and robust daily rhythms, yet are differentially responsive to altered sleep/wake and behavioral state. Given this temporal discordance, we next defined the interdependence of RR and $PR_{seg}$ measures over long (24 h) and short (5 min) time intervals (Fig. 1G–I). As expected, RR and QT interval profiles were closely aligned across the day. By contrast, the diurnal profile of $PR_{seg}$ was offset (phase delayed) from that of RR interval under normal (baseline) conditions (Fig. 1G), as confirmed by cross-correlation analyses (Fig. 1H). The relative insensitivity of $PR_{seg}$ to changes in RR was also evident over short timescales, where acute changes in RR interval between consecutive 5-min analytical bins were accompanied by a concordant change in QT interval, but not in $PR_{seg}$ (Fig. 1I). The insensitivity of $PR_{seg}$ duration to TSD and recovery nap, the phase offset between rhythms in RR and $PR_{seg}$, and the insensitivity of $PR_{seg}$ to acute changes in RR were all similarly observed in the experienced shift workers (Supplementary Fig. 3). Together, these findings reveal a pronounced difference in the regulation of SA and AV nodal function and in their response to acute shifts in behavioral routine.

We next investigated the impact of a simulated shift-work routine on cardiac electrophysiology in a second in-laboratory study (Fig. 2A–C and Supplementary Table 2[12,21]). Here, healthy individuals were assigned to either a day-shift or night-shift routine ($n = 7$/condition). As in the first study, diurnal rhythms in RR, QT, and $PR_{seg}$ intervals were evident under baseline conditions and maintained throughout the study in day-shift individuals (Fig. 2B). Following baseline ECG recording, the behavioral cycles of the night-shift group were reversed, and day/night differences in RR and QT intervals rapidly adjusted to the

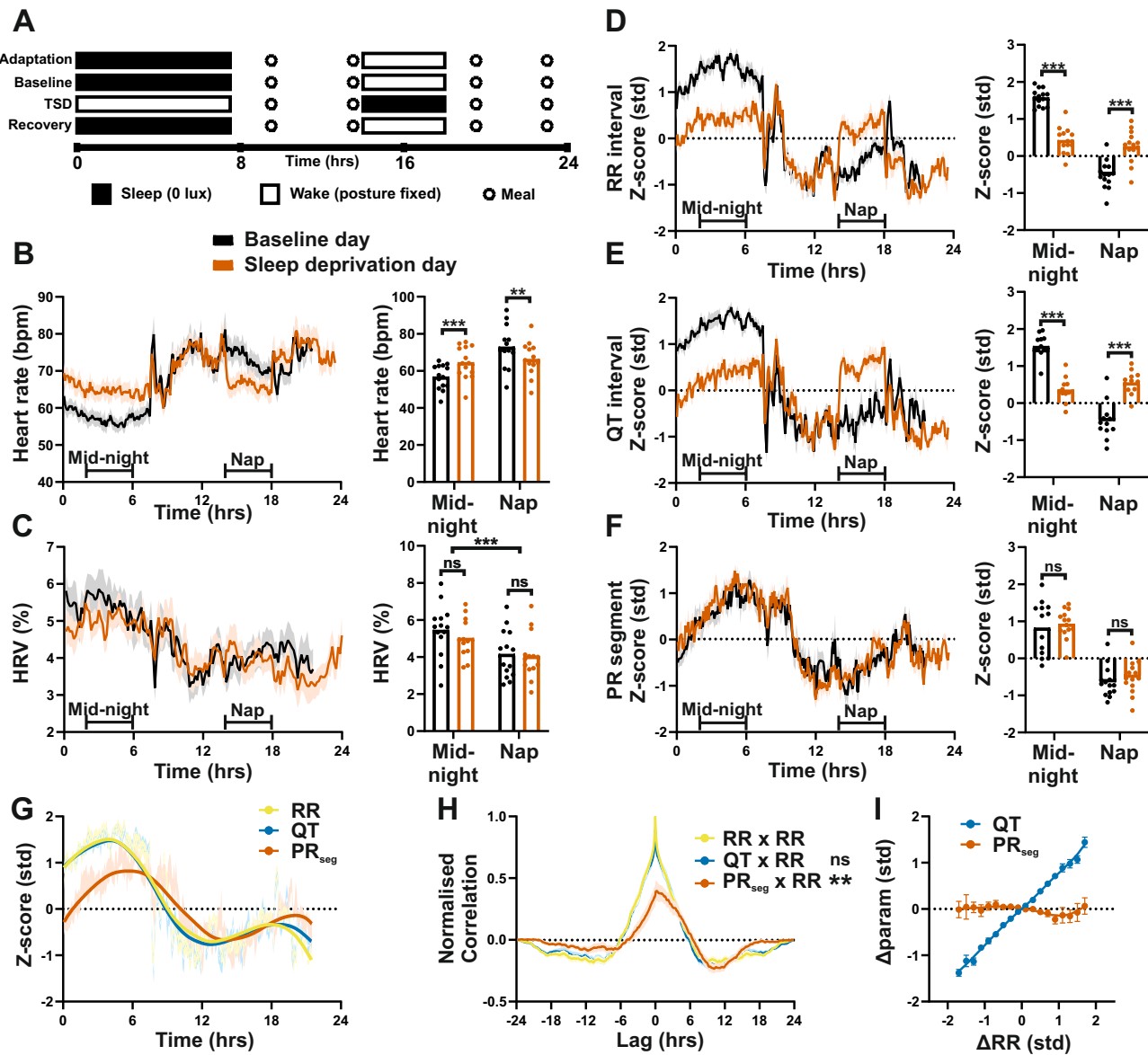

**Fig. 1 Longitudinal ECG reveals differential circadian regulation of sinoatrial (SA) and atrioventricular (AV) node function and decoupling by mistimed sleep. A**. Schematic of 4-day laboratory session[20], with total sleep deprivation (TSD) and recovery nap on day 3 (n = 14 individuals). **B–F** HR (**B**), HRV (**C**), and z-scored ECG parameter profiles under baseline (black) and TSD (orange) days (shading indicates +/−SEM), highlighting the profound impact of mistimed sleep on RR (**D**) and QT (**E**), but not PR_seg (**F**). HR has been derived from the RR interval and is shown for clarity. At baseline, all parameters were rhythmic based on cosinor analysis (P < 0.001). Individual traces were excluded from waveform analysis where data coverage fell <70% of the 5-min time bins; n (baseline/TSD) = 13/14 (**B**), 12/14 (**C**), 13/14 (**D**), 11/12 (**E**), 12/14 (**F**). Mean ECG parameters were quantified across a 4 h mid-night and mid-day (equivalent to the nap window on day 3) analysis windows on the baseline and TSD days (two-way RM ANOVA/Mixed model; n = 14 (**B, C, D, F**), 12 (**E**). **G, H** Under baseline conditions, LOWESS fit of ECG profiles (**G**) and cross-correlation (**H**; yellow: RR vs RR, blue: QT vs RR, orange: PR_seg vs RR) revealed a significant phase delay in PR_seg rhythm relative to that of RR (Gaussian fit with one-sample T test; n = 13, 11, 12, respectively). **I** Acute changes in RR were mirrored by a concordant change in QT, but not PR_seg duration (ΔRR reflects z-scored difference in RR between sequential 5-min analysis bins; Δparam reflects the concurrent change in QT or PR). All data presented as group mean ± SEM; ns P > 0.05, **P < 0.01, ***P < 0.001. bpm = beats per minute; std = standard deviation. See Supplementary Fig. S2 for additional information; source data and statistical details are provided as a Source Data File.

newly imposed schedule. By contrast, the daily rhythm in PR_seg did not adapt to the night-shift routine throughout the study (Fig. 2C). This lack of rapid adaptation in the PR_seg rhythm is consistent with slow circadian re-entrainment after an abrupt behavioral phase shift. Immediately following the 4-day-shift routine, all participants were placed under a constant routine protocol—a gold-standard procedure that removes or holds constant external timing influences (through constant dim light fixed ambient temperature, hourly isocaloric snacks) and behavioral rhythms (through constant semi-recumbent posture,

no sleep) to reveal circadian clock-controlled processes. Indeed, RR, QT, and PR_seg parameters exhibited circadian rhythmicity under constant routine in both day- and night-shift groups (Fig. 2D–F). Rhythms in RR, QT, and PR_seg exhibited a similar phase (i.e., time of peak interval duration) between day- and night-shift groups when aligned to external clock time (but not when aligned to time awake). This suggests that the circadian mechanism that underpins ECG rhythmicity is robust against acute change in behavioral patterns, and opposes the more direct influence of activity and sleep state on cardiac electrophysiology.

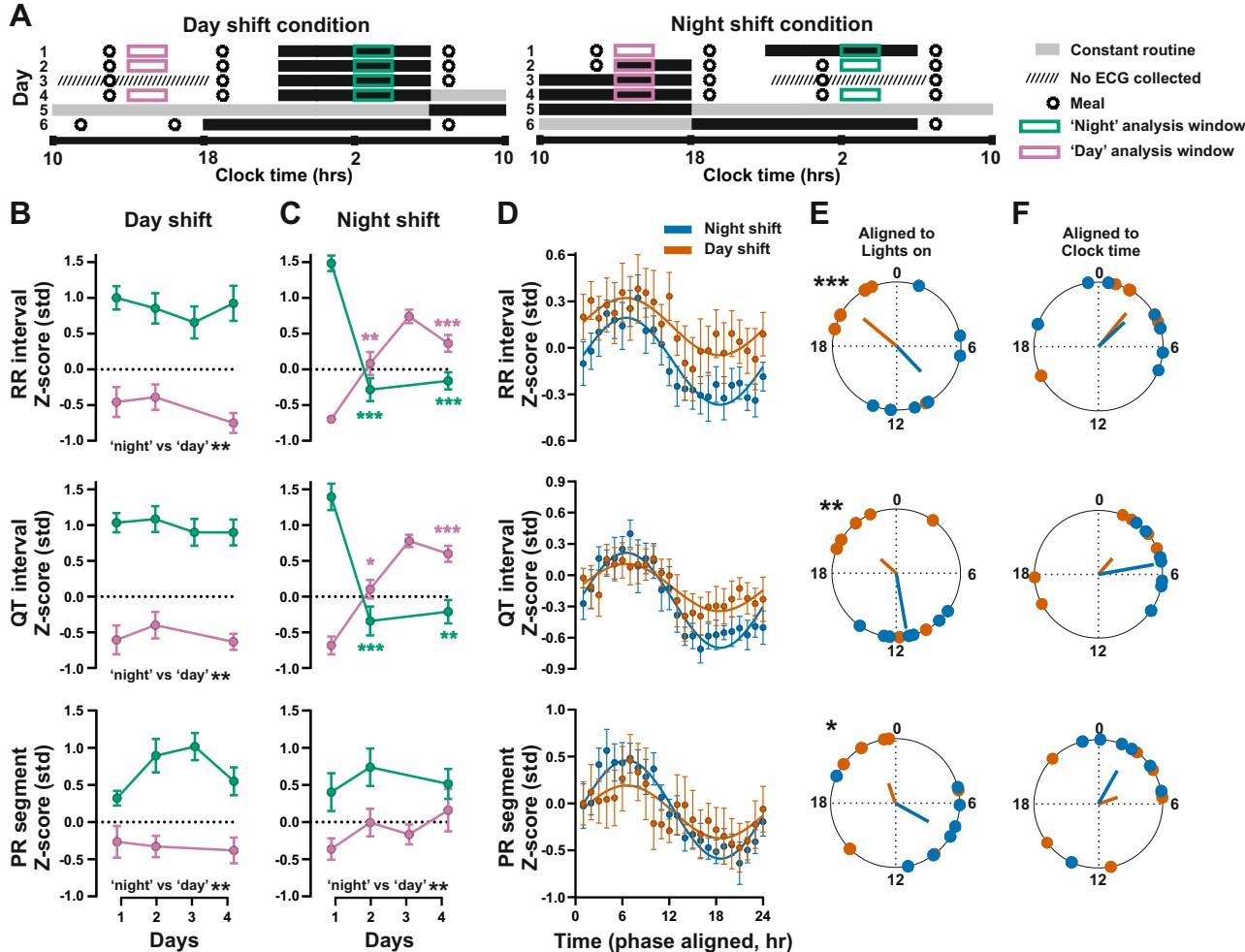

**Fig. 2 ECG response under simulated night-shift and constant routine. A** Schematic of 6-day laboratory session with simulated day-shift (left) and night-shift (right) routines (n = 7 individuals/group[21]). **B, C** ECG parameters recorded during mid-day (pink) and mid-night (green) showed a rapid reversal of RR and QT interval rhythms, but not that of $PR_{seg}$ in response to the switch to night-shift behavioral routine. Two-way ANOVA with repeated measures, colored asterisks indicate the difference from day 1. **D** Hourly binned z-scored group data and sinusoid fits from day- (orange) and night-shift (blue) conditions. Data presented as group mean ± SEM. **E, F** Timing (acrophase) of individual rhythm peaks relative to lights on (i.e., start of constant routine) (**E**) or external clock time (**F**). Asterisks indicate differences between groups (Watson–Williams test). *$P < 0.05$, **$P < 0.01$, ***$P < 0.001$. Source data and statistical details are provided as a Source Data File.

Thus, two independent human studies clearly reveal circadian rhythmicity in chronotropic and dromotropic properties of the heart. However, these inputs are not equal at the SA and AV nodes. Moreover, while the SA node is also heavily influenced by acute alterations in behavior, the AV node is robust against such changes.

**Differential circadian characteristics of SA and AV node activity are consistent between mice and humans.** To gain mechanistic insight into where and how circadian control is exerted over cardiac electrophysiology and whether it contributes to arrhythmia susceptibility, we turned to murine models. Wild-type mice studied under a 12-h light, 12-h dark (LD) cycle exhibit robust daily rhythms in HR, HRV, and ECG characteristics, including RR, QT, and $PR_{seg}$ intervals (Fig. 3A and Supplementary Fig. 4). Importantly, analysis of ECG rhythmicity in mice revealed striking similarity to the human findings detailed above, despite mice being nocturnal. Mice commonly exhibit a bout of decreased locomotor activity (and concurrent increase in sleep) in late-night (Fig. 3A, arrow)[23]. This "*siesta*" period was

accompanied by a significant decrease in HR, and lengthening of RR and QT intervals; however, parallel changes in $PR_{seg}$ duration were not evident (Fig. 3A, B and Supplementary Fig. 4B). This is akin to the response to daytime nap in human studies. To examine more directly the impact of locomotor activity on ECG parameters, we identified recording intervals showing increased activity which were then followed by >45 min of continuous inactivity (Fig. 3C). These analyses revealed a significant influence of locomotor activity on RR and QT intervals (as expected), and a much smaller yet still significant response in $PR_{seg}$ duration. Nevertheless, for all three parameters, this direct impact of activity cannot account fully for their normal diurnal variation (~2 SD from peak to trough). Isolation of sporadic activity bouts (bouts of locomotor activity within a 5-min analysis bin both preceded and followed by inactivity) revealed a significant response in RR and QT intervals but not in $PR_{seg}$ (Fig. 3D). Similar to observations in the human studies, acute changes in RR interval (between 5-min analysis windows) were reflected in concordant responses in QT, but not of $PR_{seg}$ (Fig. 3E). Thus, in mice, like humans, $PR_{seg}$ duration exhibits a robust daily rhythm but is buffered against acute changes in locomotor activity or HR.

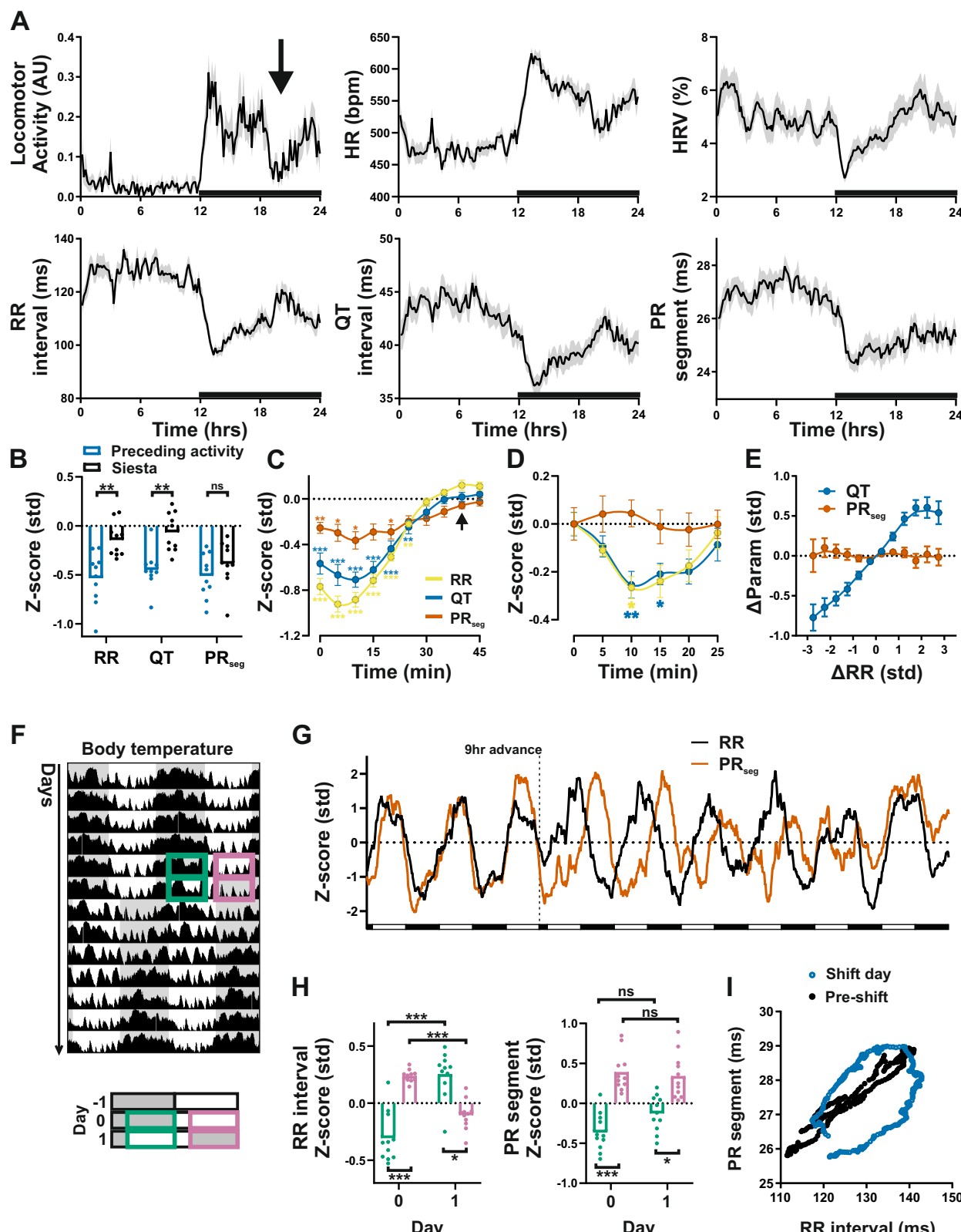

We next assessed the impact of an acute phase shift of the LD cycle in mice, thereby mimicking the mistimed sleep and simulated shift-work studies in humans. Here, mice were exposed to an abrupt 9 h advance in the LD cycle (Fig. 3F). This caused a rapid adjustment of the RR interval rhythm to the new timing (Fig. 3G, H). By contrast, rhythms in PR$_{seg}$ were much slower to adapt, leading to a temporary misalignment of RR and PR$_{seg}$

interval rhythms within individual mice (Fig. 3G) and across the population mean (Fig. 3H, I). As evident in the body temperature profile (Fig. 3F), the central circadian clock shows slow gradual adaption to the new LD schedule. That body temperature and PR$_{seg}$ interval rhythms exhibit similar shift dynamics (Supplementary Fig. 4) suggests that the AV node is similarly influenced by a central clock mechanism. Thus, despite mice and humans

**Fig. 3 Longitudinal ECG recording reveals differential circadian regulation of SA and AV node function in mice. A** Rhythms in locomotor activity (LA, arbitrary units) and ECG-derived heart rate (HR), HRV, RR, QT, and $PR_{seg}$ parameters in mice (based on 5 days of recording; shading reflects $+/-$ SEM; x axis black bar indicates dark phase; $n = 10$ mice). **B–E** Impact of LA and HR on ECG parameters. **B** Decreased LA during mid-night *siesta* (marked by an arrow in **A**) was accompanied by a significant decrease in RR and QT intervals, but not in $PR_{seg}$ duration (2-h *siesta* period (black) vs preceding 2 h of activity (blue); two-way RM ANOVA). **C** Periods of LA followed by >45 min of complete inactivity were isolated and aligned to the cessation of activity (time 0). RR (yellow), QT (blue), and to a lesser extent $PR_{seg}$ (orange) showed a significant response to the activity which decreased over subsequent inactivity (two-way RM ANOVA, Dunnet's post hoc, difference from $t = 40$). **D** Transient bouts of LA (preceded and followed by inactivity) caused a significant response in RR and QT interval lengths, but not $PR_{seg}$ (two-way RM ANOVA, Dunnet's post hoc, difference from $t = -5$). **E** Acute change in RR (across 5-min analysis bins) was mirrored by concordant changes in QT, but not $PR_{seg}$. **F** Representative body temperature profile recorded across a 9-h advance of the LD cycle (data are double plotted; shaded regions indicate periods of darkness). **G** The 9-h phase advance led to profound separation in RR (black) and $PR_{seg}$ (orange) interval rhythms. **H** Mean RR and $PR_{seg}$ intervals measured across dark (green) and light (pink) phases the day prior to shift (day 0) and equivalent times on the day of the shift (day 1; two-way RM ANOVA, $n = 11$ mice). **I** Misalignment of RR and $PR_{seg}$ rhythms in response to the shift in LD cycles disrupts the normal temporal relationship of the two parameters. Data reflect group mean $PR_{seg}$/RR on the day prior to shift (black) and the shift day (blue). All data presented as mean ± SEM. ns $P > 0.05$, *$P < 0.05$, **$P < 0.01$, ***$P < 0.001$. Source data and statistical details are provided as a Source Data File.

being night or day active (respectively), circadian control of the SA and AV nodes is strikingly similar.

The source of circadian control at the SA and AV nodes remains unclear, although both temporal variations in autonomic nervous system input and the local cardiomyocyte clock have been implicated in daily rhythms in cardiac function[3]. To discriminate these rhythmic influences, we undertook a sequential pharmacological blockade of sympathetic and parasympathetic branches of autonomic input using metoprolol and atropine (respectively). Autonomic blockade in conscious free-moving mice was confirmed by a pronounced reduction in HRV (Fig. 4A). The sympathetic blockade caused an expected lengthening of RR interval at the beginning of the animals' active phase (*zeitgeber* time 12; ZT12), yet significant time-of-day differences remained in RR, QT, and $PR_{seg}$ intervals. However, the diurnal variation in $PR_{seg}$ was lost upon complete autonomic blockade (Fig. 4B), strongly suggesting that daily rhythms in AV node conduction delay are dictated by circadian input via the parasympathetic branch of the autonomic nervous system.

By contrast to $PR_{seg}$, significant time-of-day differences in RR and QT intervals were maintained under complete autonomic blockade (reflecting intrinsic HR; Fig. 4B). This reveals a role for a local cardiac clock in setting HR rhythms across the day. To test this directly, we generated a mouse line with *Bmal1* (essential for circadian clock function) deleted selectively in cardiomyocytes ($\alpha MHC^{CRE}Bmal1^{Fl/Fl}$). Although RR and QT intervals are lengthened overall in these mice[7,9], robust daily rhythms in RR, QT, and $PR_{seg}$ are maintained (Supplementary Fig. 5). Importantly, complete autonomic blockade eliminated time-of-day differences in RR and QT intervals in the $\alpha MHC^{CRE}Bmal1^{Fl/Fl}$ but not littermate control mice, confirming that the cardiac clock contributes to setting 24-h rhythms in these parameters (Fig. 4C). To highlight the relative influence of the cardiomyocyte clock over HR, we plotted RR interval distribution across 24 h in control ($Bmal1^{Fl/Fl}$) and $\alpha MHC^{CRE}Bmal1^{Fl/Fl}$ mice (Fig. 4D). These plots suggest that local cardiac clock activity confers a time-of-day-dependent rhythm in excitability within the SA node, onto which sympathetic and parasympathetic regulation is imposed. Indeed, microdissection of the SA node across 24 h revealed not only robust clock function in this site, but also the rhythmic expression of influential ion channels and related factors (Supplementary Fig. 6). The influence of *Bmal1* over SA node pacing and ion channel expression has also recently been reported by another group, supporting an important role for the local clock in SA node function[24]. Comparison between the two genotypes also suggests that the cardiomyocyte clock modulates sensitivity to parasympathetic input at specific times of day, due to the increased prevalence of low HR across the day and late-night (Fig. 4D).

**The cardiomyocyte clock dictates a heart-intrinsic time-of-day vulnerability to ventricular arrhythmia.** To remove all systemic cues, we next recorded electrical activity directly from the atrial and ventricular myocardium of Langendorff-perfused $\alpha MHC^{CRE}Bmal1^{Fl/Fl}$ and control hearts. Similar to in vivo autonomic blockade studies, control hearts displayed a robust time-of-day difference in RR interval that was abolished in hearts lacking a functional cardiomyocyte clock (Fig. 5A). No time-of-day difference was observed in atrial to ventricular delay (time between atrial and ventricular action potentials, akin to $PR_{seg}$) in control or $\alpha MHC^{CRE}Bmal1^{Fl/Fl}$ hearts, confirming in vivo observations (Fig. 5B). This cardiomyocyte clock-governed variation in HR implies a daily variation in pacemaker cell excitability, supported by rhythmic ion channel expression in the SA node (Supplementary Fig. 6).

To examine more generally the potential contribution of the clock to cardiomyocyte excitability, we recorded spontaneous firing in isolated neonatal cardiomyocytes. This revealed a pronounced circadian rhythm in spontaneous firing frequency, with firing rhythms aligning to cell-inherent oscillations in the clock protein PERIOD2 (measured by PER2::LUC bioluminescence in parallel cardiomyocyte cultures) (Fig. 5C–E). This suggests that the propensity of pacemaker cardiomyocytes to spontaneously depolarize and fire action potentials varies across the day and that this temporal pattern is driven by a local cellular circadian clock.

Given this intrinsic time-of-day rhythm in cardiomyocyte excitability, we next investigated the susceptibility to induced arrhythmia via direct electrical stimulation in the perfused heart preparation. A strong diurnal variation in susceptibility to ventricular tachycardia (VT) was observed in hearts isolated from control animals (Fig. 5F, G). Increased propensity to arrhythmia was observed at ZT12, which is the beginning of the animals' active phase, similar to the early morning peak in the occurrence of sudden cardiac death in humans[25]. Remarkably, time-of-day variation in susceptibility to arrhythmia was not evident in $\alpha MHC^{CRE}Bmal1^{Fl/Fl}$ hearts, with these hearts being resistant to electrically induced VT. Thus, the cell-autonomous propensity for electrical firing is governed by the cardiomyocyte circadian clock, and this inherent rhythmicity is associated with time-of-day susceptibility to ventricular arrhythmias.

To confirm these findings in vivo and specifically interrogate vulnerability to adrenergic challenge, we employed an established method for catecholamine-induced arrhythmogenesis. As previously reported[26,27], wild-type mice are relatively insensitive to the administration of caffeine and adrenaline, whereas arrhythmia-prone mice, such as those carrying mutation or deletion of the ryanodine receptor (*Ryr2*) or calsequestrin (*Casq2*)

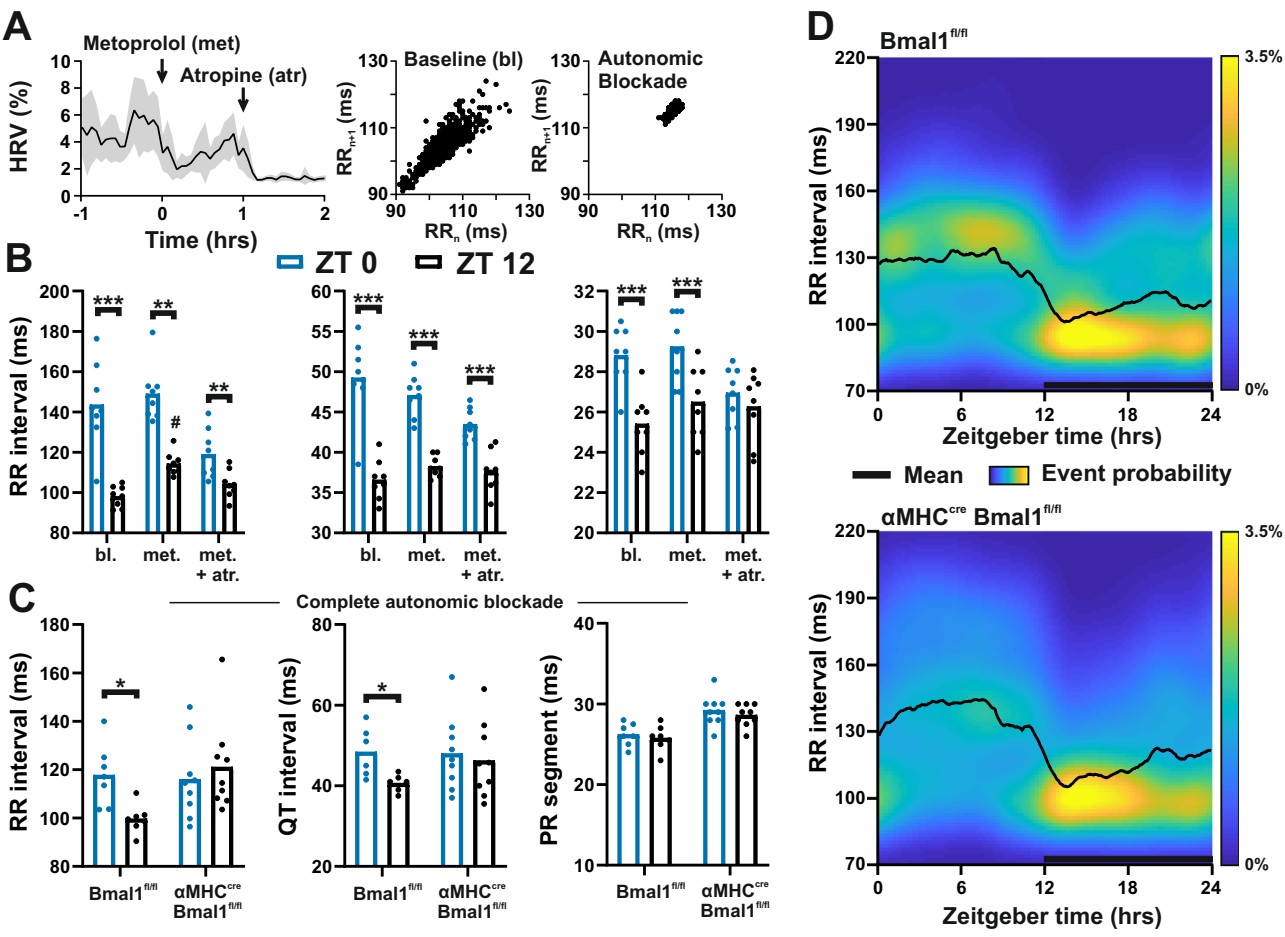

**Fig. 4 Relative contribution of autonomic nervous system input and local cardiac clock in setting RR and PR$_{seg}$ in mice. A** Heart rate variability (HRV) during autonomic blockade (with metoprolol and atropine) in conscious free-moving mice (left panel: a geometric measure of HRV, shaded region represents +/− SEM, right panels: Poincaré plots pre-blockade (baseline, bl) and under complete block). **B** Complete autonomic blockade (met + atr) reduced, but did not remove, time-of-day-dependent differences in RR or QT intervals (blue: ZT0; black: ZT12; $n = 8$ mice). In contrast, no time-of-day difference was observed in PR$_{seg}$ under complete autonomic blockade (two-way RM ANOVA, Sidak's post hoc). **C** In contrast to $Bmal1^{Fl/Fl}$ control mice, no time-of-day differences in RR or QT intervals were evident in $\alpha MHC^{CRE}Bmal1^{Fl/Fl}$ mice following complete autonomic blockade ($n = 7$ $Bmal1^{Fl/Fl}$, 9 $aMHC^{cre}Bmal1^{Fl/Fl}$; two-way RM ANOVA, Sidak's post hoc). **D** Twenty-four hours distribution of RR intervals in control ($Bmal1^{Fl/Fl}$; top) and littermate cardiomyocyte-specific $Bmal1$ knockout mice ($\alpha MHC^{CRE}Bmal1^{Fl/Fl}$; bottom). Heatmap shows the occurrence of RR interval distribution across 5 days of recording; black line reflects group mean. Substantial RR interval lengthening and variability were evident in the day and late-night. All data presented as mean ± SEM. *$P < 0.05$, **$P < 0.01$, ***$P < 0.001$ between time, #$P < 0.05$ between treatment. See Supplementary Fig. S8 for additional information related to this figure. Source data and statistical details are provided as a Source Data File.

genes, progress rapidly to bidirectional ventricular tachycardia (BVT). In line with these reports, we found that daytime (ZT0) administration of caffeine and adrenaline to conscious free-moving control mice resulted in only one of six animals tested progressing to BVT (Fig. 6A, B). Remarkably, however, catecholamine challenge at the start of the animals' active phase (ZT12) elicited BVT in all mice tested (six of six). This reveals a previously unrecognized and profound effect of time of day on arrhythmia propensity in vivo. We then challenged the $\alpha MHC^{CRE}Bmal1^{Fl/Fl}$ mice at this vulnerable phase (ZT12) and found an attenuated response, with only two of five mice showing BVT (Fig. 6B and Supplementary Fig. 7A).

It is likely that increased vulnerability to catecholamine-induced VT is associated with increased cardiomyocyte excitability at night. To define potential mechanisms underlying both the diurnal rhythms in arrhythmogenesis and the relative protection obtained from cardiomyocyte-specific $Bmal1$ deletion, we profiled ventricular gene expression in control and $\alpha MHC$-$^{CRE}Bmal1^{Fl/Fl}$ mice at ZT0 and ZT12 (Supplementary Fig. 7B).

Differential gene expression revealed a number of candidate ion channels and other factors (e.g., connexins) that have been linked to ventricular arrhythmia. Across both time of day and genotype, genes linked to intracellular calcium handling were particularly evident, including significantly increased expression of $Casq2$ and decreased expression of $Ryr2$ and $Cacna1c$ (Ca$V_{1.2}$ subunit of the L-type calcium channel) in $\alpha MHC^{CRE}Bmal1^{Fl/Fl}$ animals, relative to control mouse hearts. These clock-directed changes in calcium handling are likely to underpin the relative sensitivity to induced arrhythmias and highlight a potential avenue for future therapeutic intervention.

## Discussion

These findings provide a critical understanding of how circadian, autonomic, and behavioral inputs interact to determine the temporal dynamics of cardiac electrophysiology and time-of-day susceptibility to arrhythmia. We reveal a functional segregation of acute (e.g., reflective of activity and/or sleep state) and circadian inputs from the autonomic nervous system and local

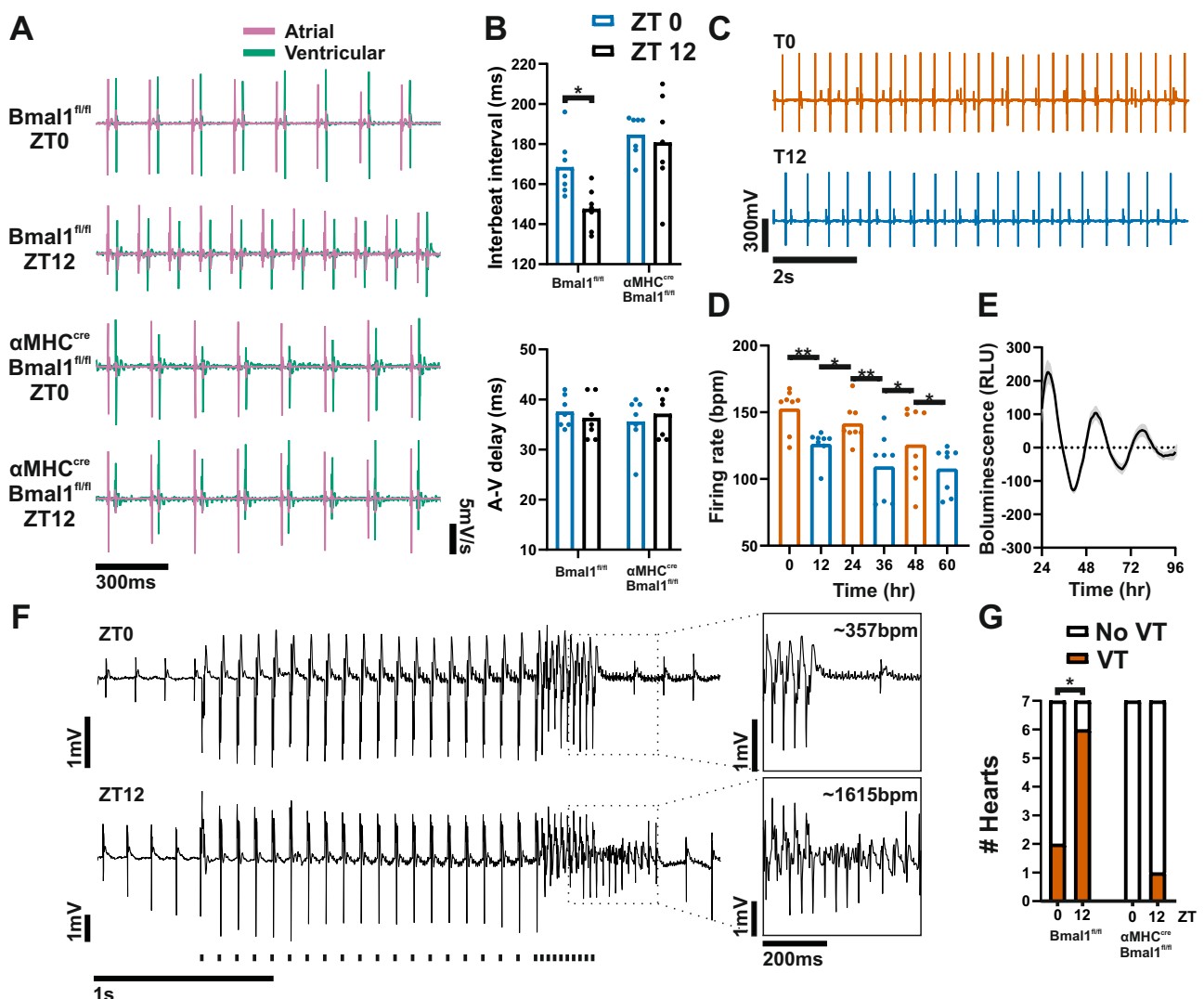

**Fig. 5 A local heart clock drives time-of-day-dependent rhythms in cardiomyocyte excitability. A** Representative recordings of spontaneous atrial (pink) and ventricular (green) electrograms recorded in ex vivo Langendorff-perfused hearts collected from control (*Bmal1^{Fl/Fl}*) and *αMHC^{CRE}Bmal1^{Fl/Fl}* mice at ZT0 or ZT12. **B** Interbeat interval showed a time-of-day (ZT0 blue, ZT12 black) difference in control but not *αMHC^{CRE}Bmal1^{Fl/Fl}* hearts; A–V conduction delay in isolated hearts did not vary by time of day (*n* = 7 hearts/group, two-way RM ANOVA, Sidak's post hoc). **C** Representative multielectrode array recording of spontaneous action potential firing in primary cardiomyocytes. **D**, **E** Spontaneous firing rate in isolated cardiomyocytes showed robust circadian variation in constant culture conditions (**D**, *n* = 8, one-way RM ANOVA, Holm–Sidak post hoc), which followed rhythms of mPER2::LUC bioluminescence recorded in parallel cultures (RLU: relative light units, **E**). **F** Representative ventricular traces and pacing protocol (stimulation train shown below recording). Inset showing recovery to normal sinus rhythm (top) and induced ventricular tachycardia (VT; bottom). **G** Control hearts (*n* = 7/timepoint) showed a significant time-of-day susceptibility to VT (ZT0: 2 of 7 hearts tested, ZT12: 6 of 7; Chi-square), whereas only 1 of 14 *αMHC^{CRE}Bmal1^{Fl/Fl}* hearts (*n* = 7/timepoint) were susceptible. All data presented as mean ± SEM. *P < 0.05, **P < 0.01. Source data and statistical details are provided as a Source Data File.

cardiomyocyte clock influences over SA and AV nodal activity. This segregated input reduces coordination within the conduction system in response to acute shifts in the timing of the behavioral routine. Importantly, these findings appear robust across both human and mouse physiology. Our work also reinforces the importance of daily rhythms in parasympathetic tone in shaping circadian variation in HR and HRV and extends this to include a profound influence over rhythms in AV node conduction. There is evidence that vagal innervation of the SA and AV nodes can be segregated at the level of the cardiac ganglion and medulla nuclei[28,29], and our findings suggest that these pathways may also differentially deliver circadian input (Supplementary Fig. 8). Indeed, the multisynaptic autonomic connection from the master SCN clock to the heart has been demonstrated[5], and we have

recently shown that the activity of VIP-expressing SCN neurons can influence HR in a time-of-day-dependent manner[30].

An influence of the local circadian clock on cardiac function is already established in rodent models, with the cardiomyocyte clock exerting rhythmic regulation over ~10% of the heart transcriptome[7,31,32]. We extend these findings to the SA node, where the inherent time-of-day-dependent rhythms in sinus node pacing were evident in vivo, under complete autonomic blockade, and in isolated hearts. We also show circadian clock function and rhythmic expression of influential ion channels within the SA node, providing a mechanism for daily rhythms in inherent HR. These findings are in line with previous work showing an important influence of the heart clock on heart contractility, myocardial metabolism, transcriptional activity, and protein

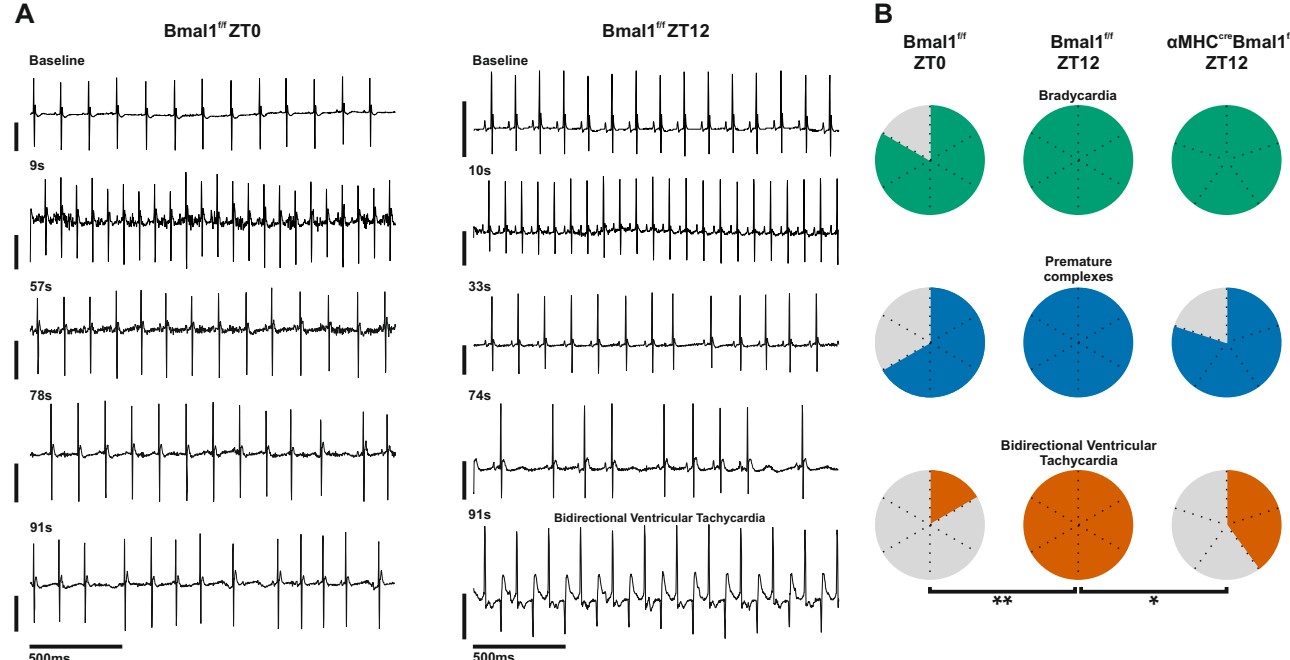

**Fig. 6 In vivo propensity for catecholamine-induced bidirectional VT displays diurnal rhythmicity. A** Typical ECG traces before (baseline) and after injection of caffeine and adrenaline. Animals displayed rapid tachycardia, typically followed by bradycardia and sinus pauses and/or premature complexes, with some progressing into bidirectional VT (bottom right). Vertical scale bars represent 0.5 mV and inset times are time from returning to the cage after injection. **B** Proportion of animals that display evidence of each stage of VT progression. At ZT12, all control animals (six of six mice tested) displayed robust evidence of bidirectional VT, compared to only one of six mice tested at ZT0. Only two of five αMHC^CREBmal1^Fl/Fl mice displayed bidirectional VT at ZT12. *$P < 0.05$, **$P < 0.01$, Chi-square. Source data and statistical details are provided as a Source Data File.

synthesis turnover[1,33]. Indeed, altered expression of cardiac ion channels (e.g., *Scn5a*, *Kcnh2*) and some electrophysiological measures (HR, QT interval) have been described in cardiomyocyte-specific clock disrupted mice[4,7,9,34,35]. Here, we also show explicitly that the cardiomyocyte clock dictates temporal rhythms in cell excitability and susceptibility to ventricular arrhythmia.

Previous studies have shown that wild-type mice are relatively resistant to catecholamine-induced arrhythmias[26,27]. This is in line with our results from the animals' rest phase. However, a dramatic increase in progression to BVT was observed at night, similar to that previously ascribed to responses observed in susceptible transgenic mouse lines. Significant evidence in animals and humans has shown that disturbances in intracellular calcium handling can increase susceptibility to VT. In response to myocyte depolarization, L-type calcium channels activate and promote calcium influx into the cytoplasm. This calcium influx activates RYR2 on the sarcoplasmic reticulum (SR), leading to a further substantial release of calcium ions from the SR, which in turn activates myofilament activation and initiates contraction. CASQ2 serves to bind calcium ions within the SR, thereby limiting this intracellular calcium release. Reduction in CASQ2 levels and/or mutations of *Casq2* can cause arrhythmogenic calcium release, leading to premature ventricular contractions and arrhythmia. Conversely, overexpression of *Casq2* can have a protective effect due to increased calcium retention[36,37]. In hearts lacking *Bmal1* expression in cardiomyocytes, we found downregulation of *Cacna1c* and *Ryr2*, and marked upregulation of *Casq2*. Thus, a decreased propensity to intracellular calcium overload likely underlies the relative protection afforded by the cardiomyocyte deletion of *Bmal1*.

The cardiomyocyte clock has previously been linked to cardiac arrhythmias. For example, the genetic deletion or overexpression of the transcriptional regulator KLF15 in mice can lead to abnormal cardiac repolarization and enhanced susceptibility to ventricular arrhythmias[10]. KLF15 is regulated by *Bmal1* in the heart, suggesting that clock-controlled rhythmicity in its expression could contribute to time-of-day variation in arrhythmogenesis. We find a robust time-of-day difference in Klf15 expression in ventricular tissue of control animals and significant downregulation in cardiomyocyte-specific *Bmal1* knockouts. In addition, we identify rhythms in the expression of genes from a range of potential systems that have been linked to ventricular arrhythmias in human and mouse studies, including *Calm2*, *Kcnh2*, *Scn5a*, *Cx43*, and *Cx45*[38–42]. While it is unlikely that any one of these gene changes alone is driving the temporal pattern of arrhythmias, our data suggest a generally more excitable state within the ventricle at ZT12, driven in part by the cardiomyocyte clock.

We believe that there are additional aspects of cellular rhythmicity, beyond oscillations in gene expression that contribute to cardiomyocyte electrical state. Our recent work has revealed a fundamental role for daily oscillations in cytosolic protein concentration and temporal variation in K$^+$ and Na$^+$ gradients in dictating cardiomyocyte electrical state across the day[43]. This work suggests that cardiomyocyte clock organization of the daily timing of mTORC activity and macromolecular crowding drives a concurrent oscillation in cytosolic ion concentration, which imparts cell-intrinsic daily variation to firing frequency. Indeed, increased mTOR expression and elevated protein synthesis have been reported in cardiac *Bmal1* knockout mice[44], supporting the view that alterations in overall protein synthesis contribute to the electrophysiological phenotype in our mice.

It is clear that long-term shift work is associated with an elevated risk of cardiovascular disease, the incidence of cardiac events, and altered electrophysiological parameters[15,45–48]. Whether alteration of cardiac conduction parameters during mistimed sleep and shift-work routines increases susceptibility to

arrhythmia or other harmful cardiac events in otherwise healthy humans is not yet clear. Nevertheless, it is likely to be of important clinical consideration in patients with pre-existing cardiac dysfunction or injury, as well as in relation to ECG-based diagnoses and pharmacological intervention, where the time of day, patient occupation, and/or sleep–wake history may significantly impact the outcome. Moreover, we show that susceptibility to VT was rarely observed upon cardiomyocyte *Bmal1* deletion, indicating that the circadian clock drives increased excitability during the active period of the day at the cost of creating vulnerability to arrhythmias. This suggests that clock function within the heart contributes to the long-known temporal variation in cardiac arrhythmia propensity observed in humans. Given the widespread influence of the circadian clock and established detrimental consequences of clock disruption, any clock-directed intervention must be approached with caution. Nevertheless, our findings offer an important and logical new avenue for therapeutic investigation.

## Methods
### Human laboratory studies

*Ethics.* The human research studies complied with all the current ethical requirements and guidelines of the University of Surrey and Washington State University, and conformed with the Declaration of Helsinki (2004), the Universal Declaration of Human Rights and Covenants on Human Rights (UN General Assembly, December 1948) and Data Protection Act, 1998. Study protocols were approved by their respective ethics committees at the University of Surrey and Washington State University and complied with all aforementioned ethical regulations. Participants gave written, informed consent and met predefined inclusion criteria[20,21].

*Study one.* Full details of this study are described in refs. [20,49]. Briefly, 11 long-term shift workers (males, aged 25–45 years) with a shift-work history of >5 years and 14 control participants (males, aged 25–42 years) were asked to keep a regular 7.5–8-h sleep period prior to the laboratory session. During the laboratory session, volunteers were kept in tightly controlled conditions for 4 days with self-selected sleep durations of 7.5–8 h and wake times ranging from 5:30 to 8:00 h. All analyses were done relative to the participants' wake time. After a day of adaptation and baseline recording, participants were kept awake for 30.5 h and allowed a subsequent 4-h recovery nap (Fig. 1A). This was followed by a normal recovery day. On control days (1, 2, and 4), subjects were posture fixed in a semi-recumbent position for the same 4 h window as the nap on day 3. The light was fixed at a dim ~8 lux throughout wake periods and 0 lux during sleep periods. ECG was recorded at 256 Hz throughout using a wireless polysomnographic system (Siesta EEG/PSG recorder, Compumedics Ltd, Australia) via electrodes in two positions (right midclavicular and around 6 cm under the left armpit) using ProFusion PSG2 (version 2.1, Compumedics Ltd). Data were converted to the European data format for further analysis.

*Study two.* Full details of this study are described in refs. [12,21]. Briefly, 14 participants (10 men, 4 women, aged 22–34 years) were randomly assigned to either the "day" or "night" shift conditions. Prior to the laboratory session, participants maintained a self-selected regular sleep schedule. During the laboratory session, participants had fixed, 8-h windows of sleep opportunity. Both groups had 1 day of baseline recording with sleep between 22:00 and 06:00 h before the "night shift" group were, after a transition nap between 14:00 and 18:00 h, shifted to a 3-day period with a sleep window between 10:00 and 18:00 h. This was followed by a day in constant routine and a final recovery day. Lights were kept below 50 lux during wakefulness. ECG was recorded at 4 kHz throughout except for an ~8-h window on the 3rd day using a Holter monitor (DMS 300-3 A; Bravo, Huntington Beach, CA, USA) with standard 5-lead electrode placement. Data were downsampled to 128 Hz for analysis.

*Pre-study conditions for both human studies.* Participants were free of any medical conditions and not currently on medication known to affect cardiovascular, metabolic, gastrointestinal, or immune function (including over-the-counter medication). Participants did not suffer from sleep, depression, or anxiety disorders as confirmed by self-reported pre-study questionnaires (both studies), history, physical exam, and baseline polysomnography (study 2). All participants were nonsmokers at the time of the study and had no history of alcohol abuse. All participants had not been involved in shift work in the previous 3 months and had not traveled across time zones in the previous month. A urine screen confirmed all subjects were negative for drugs of abuse at the time of recruitment and during the study. Participants were asked to refrain from heavy exercise (2 days prior to study 1), and substances including alcohol and caffeine (2 days prior to study 1, 1 week prior to study 2).

*Animals.* All animal experiments were licensed under the Animals (Scientific Procedures) Act of 1986 (UK) and were approved by the animal welfare committees at the University of Manchester and Laboratory of Molecular Biology, Cambridge. C57BL6J mice were purchased from Charles River (UK) and other mouse lines ($\alpha MHC:cre$[50], $Bmal1^{f/f}$[51], $\alpha MHC:cre^{+/-}Bmal1^{f/f}$) were bred at the University of Manchester. Mice imported from external breeders were acclimatized to the local animal unit for at least 7 days prior to use. Mice were housed under a 12:12 h light/dark cycle at ~400 lux during the light phase and 0 lux during the dark phase. Ambient temperature was kept at $22 \pm 2$ °C, the humidity was ~$52 \pm 7\%$, with food and water available ad libitum. Mice were group-housed until the start of ECG recording, during which they were individually housed and kept in light-tight cabinets. Male and female mice (10–16 weeks of age, ~1:1 ratio) were used unless stated otherwise.

*ECG telemetry implantation.* Mice were implanted with ETA-F20 telemetry devices (Data Sciences International, USA) for the recording of body temperature, locomotor activity, and electrocardiography. Mice were anesthetized with isoflurane (2–5% in oxygen, total duration ~20 min), following which the telemetry remote was inserted into the abdominal cavity. Biopotential leads were secured ~1 cm right of midline at upper chest level (negative) and ~1 cm left of the midline at the xiphoid plexus (positive). Mice were allowed a recovery period of 7–10 days before individual housing and the start of the experiment. For long-term recording body temperature, locomotor activity, and a 10 s ECG sweep were recorded every 5 min. For autonomic blockade studies, ECG was recorded continuously.

*ECG analysis.* All ECG analysis was done using bespoke ECG analysis software written in MATLAB (R2018a; Mathworks, USA)[11]. Briefly, sweeps of 10,000 data points were taken, corresponding to 10 s in mice and ~39//78 s in humans (analyzed at 256/128 Hz), and R waves detected by amplitude thresholding and subsequent template matching to the mean beat waveform of that sweep. Any waveforms that differed significantly from that template were removed. Individual recording sweeps were excluded from subsequent analysis where the mean amplitude of discriminated beat waveforms was <3 times the lower limit of the amplitude window, where baseline variation in the ECG trace exceeds 1/3 of that lower limit and/or where >20% of the events detected were excluded by the template matching algorithm.

For each valid beat, ECG features were automatically detected. The P wave was identified as the maximal point between the previous beat and the current R wave. P offset was set as the first positive deflection to the right of P peak in mice, and as the point at which the ECG crossed the isoelectric line in humans. Q onset was set as the first positive deflection to the left of R peak in mice, and the first negative deflection to the left of that positive deflection in humans. T peak was identified as the maximum (in humans) or minimum (in mice) point between the current R wave and next P wave. T offset was set as the point at which the ECG returns to the isoelectric line after T peak in mice, or as the first positive deflection after crossing the isoelectric line in humans, dependent on T wave morphology.

ECG parameters were then calculated for each valid beat in the sweep and the median was taken for the sweep. HR was calculated directly from the inverse of the RR interval for each sweep. Importantly, there was no day/night bias to the number of sweeps failing quality control, and adjustment of the quality control parameters had little impact on the overall results. Human ECG data were subsequently binned into 5-min bins for longitudinal analysis. In study 1, QT was not measured for four individuals (two in the control group, two in shift-work group) as the end of the T wave could not be reliably determined.

Random sweeps from each individual were visually inspected for accuracy of automated feature detection and measurement. To validate the algorithm further, we compared automatically detected PR, QT, and RR intervals of a representative sweep from control individuals with manual measurements provided by an experienced cardiologist (LAV) of the average of that sweep and found no significant differences (Supplementary Fig S1C).

Common measures of HRV have limitations. Time-domain methods, such as RR interval standard deviation (SDNN), are heavily influenced by absolute and overall changes in HR. Frequency-domain measures require interpolation from clean, stable ECGs from stationary participants, making them unsuitable for continuous, long-term ECG analysis. For these reasons, we employed a simple, robust geometric method based on relative RR intervals which is insensitive to outliers and changes in absolute HR[22].

*Autonomic blockade.* Conscious free-moving mice were injected with metoprolol (10 mg/kg, i.p., Sigma-Aldrich, UK) to achieve sympathetic autonomic blockade. In order to achieve total autonomic blockade, atropine (4 mg/kg, i.p., Sigma-Aldrich, UK) was given 40–60 min after metoprolol. These times were chosen based on when HR reaches a plateau following metoprolol injection. The total blockade was marked by a stable attenuation in HRV, with analyses conducted over the following 30 min.

*In vivo arrhythmia induction.* Mice were implanted with telemetry devices (as above) and allowed to recover for 10 days post surgery. At ZT0 or ZT12 on the day of study, age-matched male mice were injected with 120 mg/kg caffeine (Fisher Scientific, USA) and 2 mg/kg adrenaline (Sigma, US) in PBS. Animals were

monitored for 15 min before culling. Trains of at least four alternating ventricular premature complexes (characterized by a widening of QRS complexes and QRS inversion) were identified as bidirectional VT. In most cases, they lasted >30 s.

*Langendorff heart electrophysiology.* Hearts were rapidly removed and placed into ice-cold Krebs–Henseleit Buffer (KHB) solution (constituents in mM: 118 NaCl, 11 glucose, 1.8 CaCl₂, 4.7 KCl, 25 NaHCO₃, 1.2 MgSO₄, 1.2 KH₂PO₄). Excess tissue was removed, aorta cannulated under a dissection microscope and heart perfused with KHB. Successful perfusion was confirmed when all visible vessels in the heart cleared with perfusate. The cannulated heart was then attached to a heated glass coil filled with oxygenated KHB and perfused at 37 °C at 4 ml/min. This whole process took ~3–4 min. Hearts were then left to stabilize for 10 min; any hearts that did not stabilize within this window were excluded (<10%). Monophasic action potentials were recorded from the left atrial appendage and ventricular myocardium using custom-made silver chloride electrodes, amplified using Bioamplifiers, digitized by Powerlab (4/35), and analyzed in LabChart (v8; ADInstruments, UK). Electrical stimulation was applied via a custom-made silver chloride electrode to the ventricular myocardium. Electrode placement was consistent between hearts and based on anatomical landmarks. Ventricular arrhythmias were induced using an established protocol[52] and ventricular tachycardia (VT) was characterized according to the Lambeth convention[53]. Briefly, an S1 train consisting of 20 pulses at 100-ms cycle length was immediately followed by S2–S10 train of extra stimuli. Extrastimuli cycle lengths ranged from 90 down to 18 ms, decreasing in 3 ms intervals. Each segment of stimulation was separated by 2 s. VT was defined as a train of at least four consecutive ventricular premature beats following this stimulation. This process was repeated twice and hearts that displayed evidence of VT on both trials were classified as susceptible to VT. Animals were culled between ZT 0–1 and ZT 11–12.

*Cardiomyocyte isolation and recording.* Primary cardiomyocytes were isolated from P2-3 neonatal mice using a commercially available dissociation kit (130-098-373, Miltenyi Biotec, Germany). Briefly, hearts were removed and cut into small sections in ice-cold dissociation buffer (constituents in mM: 106 NaCl, 20 HEPES, 0.8 NaH₂PO₄, 5.3 KCl, 0.4 MgS0₄, 5 glucose). Following enzymatic digestion, the solution was filtered through a 70-μm filter and placed in dishes for 2 h. The cardiomyocyte-containing supernatant was collected and $1 \times 10^5$ cardiomyocytes were seeded onto the center of a multielectrode array (MEA) coated with 10 μg/ml fibronectin (Sigma). After 5 h, DMEM high glucose media (supplemented with 17% M199 (ThermoFisher, USA), 10% horse serum, 5% newborn calf serum, Glutamax, Pen/Strep, and Mycozap (Lonza)) was added. Media was then changed daily for 8 days before a final media change and release into constant 37 °C conditions. Every 12-h field potentials were recorded from the MEA using a MEA recording device (MEA2100-2 × 60-system, MCS) for 5 mins at 20 kHz following a 2-min equilibration period. The recording was done at 37 °C, at atmospheric conditions in a medium supplemented with 10 mM HEPES. Data were recorded and analyzed using the Multi-Channel experimenter and DataManager software. Data are presented from a single MEA (and replicated across different MEAs). Cardiomyocyte bioluminescence was captured using ALLIGATORs (Cairn Research) for 29 mins at 30 min intervals in constant, 37 °C conditions. Data were detrended by subtraction of the 24 h moving average.

*Gene expression analysis.* For SA node dissection, male C57B6J mice (12 weeks of age) were housed under a stable 12 h:12 h LD cycle for >10 days prior to dissection. SA nodal tissue was quickly microdissected and snap-frozen at six equal time points across 24 h. For ventricular gene expression analyses, control (Bmal1ᶠˡ/ᶠˡ) and αMHC:cre⁺/⁻Bmal1ᶠ/ᶠ mice were sacrificed at ZT0 or ZT12. In all cases, tissue was homogenized (Bead mill 24; Fisherbrand), and total RNA extracted using columns according to the manufacturer's protocol (Reliaprep tissue; Promega). cDNA was synthesized using High capacity RNA to cDNA kit (Applied Biosystems). Quantitative PCR was performed on a 384 well thermal cycler (7900; Applied Biosystems) using GoTaq Master Mix (Promega). Relative gene expression was calculated using the ΔΔCt method using Hprt, Ppib, and Actb as housekeeping genes. Primer sequences are listed as Supplementary Material (Supplementary Table S3).

*Data processing and analysis.* Experimental design and *n* number determination were based on appropriate power analysis and previous experience. All ECG data were processed in MATLAB (R2018a; Mathworks, USA) using custom-written algorithms as described below. In both human studies, data were z-scored across the full laboratory session for each individual for analysis to normalize interindividual variability. For mouse studies, data were z-scored across the whole analysis window (5 days for most analyses, ~9 days for phase shift experiments). Phase delays between RR and other ECG parameters were calculated by comparing the mean of a Gaussian fit to the cross correlogram in each individual to 0 using a one-sample *T* test. To calculate the dependency of ECG parameters on HR, change in z-scored RR between sequential 5 min ECG analysis windows were determined and segregated into bins reflecting response magnitude (width 0.25 SD in humans, 0.5 SD in mice). Corresponding changes in QT and PR_seg were averaged and plotted against ΔRR interval. To determine the effects of locomotor activity on ECG parameters a circadian profile of each parameter at rest was calculated for

each animal by fitting a 24-h constrained sine wave to all data points in which the animal had not been active in the preceding 30 min over 5 days of normal recording. Periods in which bouts of locomotor activity were followed by 45 min of inactivity were then isolated and normalized to the appropriate circadian phase based on the at-rest curve fit. The transient activity was identified as an activity bout preceded by >5 min of inactivity and followed by >20 min of inactivity. Data were normalized to the preceding 5 min of inactivity. RR interval distribution heatmaps were calculated by binning RR intervals of mice across 5 days of ECG recording into square bins of size 15 min × ~1.5 ms. Data were converted to a probability of an RR interval falling into any given bin within each 15 min window. Heatmaps were then filtered using a 5 SD Gaussian window with circular padding.

*Statistics and reproducibility.* Statistics analyses were done in Prism (v8; Graphpad, USA) using statistical tests described in figure legends. All statistical comparisons were two-sided and exact *P* values test statistics are provided in a Source Data file. Figures were compiled in CorelDraw (2018; Corel Corporation, Canada). The experimental design of in vivo animal studies was based on power analyses and previous experience. These studies were therefore not directly repeated; nevertheless, many findings were confirmed in independent studies (e.g., time-of-day response in ECG parameter under autonomic blockade) involving wild-type and transgenic mouse lines. In all studies, the reported *n* number reflects independent biological replicates (i.e., human, mouse).

**Reporting summary.** Further information on research design is available in the Nature Research Reporting Summary linked to this article.

## Data availability
The materials used and datasets generated during this study are available from the corresponding author on reasonable request. Raw ECG recordings have not been included in the Source data file and are available from the corresponding author on reasonable request. Source data are provided with this paper.

## Code availability
The ECG analysis code used in this study is available in the GitHub repository (https://github.com/EdHayter/Hayter-et-al.-ECG-analysis)[54].

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

## Acknowledgements

We would like to thank the Biological Services Units at the University of Manchester and Laboratory of Molecular Biology for their excellent support of the animal work. We also thank our funders for their generous support: UK Biotechnology and Biological Sciences Research Council (BBSRC) grants to D.A.B. (BB/J017744/1; BB/I018654/1); AstraZeneca Blue Skies Initiative and the Medical Research Council (MC_UP_1201/4) to J.S.O.; Human study one was supported by an EU Marie Curie Research Training Network grant (CT-2004-512362) to D.J.S., 6th Framework project EUCLOCK (018741) to D.J.S. and Stockgrand Ltd (University of Surrey) to S.M.T.W. and D.J.S. Human study two was supported by start-up funds from the College of Pharmacy and Pharmaceutical Sciences at Washington State University to S.G. and by Congressionally Directed Medical Research Program and U.S. Army Medical Research and Development Command awards W81XWH-16-1-0319, W81XWH-18-1-0100 and W81XWH-20-1-0442 to H.P. A.V.D.; National Institutes of Health Grants R01ES030113 and R21CA227381 to S.G.; BBSRC Grant BB/I019405/1 to D.J.S.; and EU FP7-HEALTH-2011 EuRhythDia Grant 278397 to D.J.S.

## Author contributions

Conceptualization: E.H., T.B., D.J.S., A.W.T., and D.A.B.; methodology and software: E.H. and T.B.; investigation: E.H., S.W., A.S., E.C., N.B., J.S.O., and D.A.B.; formal analyses: E.H., L.A.V., T.B., A.W.T., and D.A.B.; writing: all authors; design and supervision of human trials: H.P.A.V.D., S.G., and D.J.S.; funding acquisition: D.A.B., H.P.A. V.D., S.G., and D.J.S.

## Competing interests

The authors declare no competing interests.
