## [Peer Review File · Nature Communications]

REVIEWER COMMENTS

Reviewer #1 (Remarks to the Author):

“Distinct circadian mechanism govern rhythms in cardiac electrophysiology and susceptibility to dysrhythmia” by Hayter and colleagues works to determine how circadian clocks in the heart and brain set the rhythms in the sinoatrial node (SAN) and the atrioventricular node (AVN). They found using a combination of human and mouse studies- circadian inputs to the SAN and AVN are uncoupled during circadian misalignment by behavioral interventions. They use pharmacological approaches to show the different roles of the autonomic nervous system and intrinsic cardiac clock have on day-night rhythms in PRseg and RR intervals. Importantly, disruption of the circadian clock in the heart appears to decrease susceptibility to tachycardia in mice. Conceptually this is an interesting study. The strengths are the combined approaches to show relevance to humans and possible mechanisms in mice. Some questions and clarifications are needed.

The authors suggest they are assessing SAN activity by measuring the RR interval (the interval between ventricular depolarization). Can the authors include the PP interval in their analysis? How does the PP interval change in the longitudinal human and mouse ECG studies? Is it the same as the RR interval?

Figure 3 shows longitudinal changes in ECG recordings. How are the RR and HR data different? Is the HR calculated using RR or is the probe actually detecting HR by another method? Since this study is focused on SAN and AVN what is the purpose of showing the QT interval?

Figure 3A. Does the daily change body temperature better correlate with changes in activity similar to RR or is it similar to the PRseg? Based on Figure 3F and 3G body temperature seems to change to the new light cycle similar to PRseg. – is this true? Can you show this more directly?

Figure 5B-5E are confusing. These are neonatal cardiomyocytes. Are the authors suggesting the intrinsic electrical activity in the cells is analogous to the node? Please clarify or justify this assertion. Have the authors tried a similar study using the alpha myosin heavy chain Bmal1f1 mouse neonatal cardiomyocytes?

Figure 5F – is this representative? VT is defined as 4 consecutive premature beats following stimulation. How long did the VT normally last after stimulation in the hearts? Which animal was this from?

Are there any obvious structural differences in the hearts (size or fibrosis) of *Bmal1^{fl/fl}* and alpha myosin heavy chain *Bmal1^{fl/fl}*?

Reviewer #2 (Remarks to the Author):

The authors performed studies in humans and mice. The autonomic nervous system was perturbed using sleep deprivation in humans and drugs (metoprolol and atropine) in mice. The authors concluded that "Our findings reveal a functional segregation of circadian control across the heart's conduction system and inherent susceptibility to dysrhythmia."

There are several major methodological issues related to this work. The authors measured a single lead ECG. The P, R and QT intervals were automatically detected by a computer algorithm. There are limitations. First, the PR interval is generally measured from the onset of P wave to the onset of QRS complex. It is used to represent the time needed to propagate the impulse from sinus node through AV node and reach the ventricle. However, the PR segment used by the authors measured from the end of the P wave to sometime after the onset QRS complex, when ventricular depolarization has already occurred. The end of P wave is dependent on the completion of the atrial depolarization. Generally the left atria and/or pulmonary veins are depolarized last. The end of P wave is not determined by the AV nodal conduction. Rather, it depends on the impulse propagation within the atria. The end point of the PR segment was measured at the negative deflection of the QRS complex, when the events in the AV node has already ended. It is not a good estimation of the time when the impulse first crossed the AV node. Finally, the onset of QRS complex cannot be determined by a single lead ECG. Depending on the electrical axis, some ECG leads would record earlier onset of the QRS complex than others. That earliest onset should be used to close the PR segment.

The authors did not explain how the QT interval was measured. The "T off" in supplemental figure 1, Panel A, shows it is the time when T wave crosses the dotted line. However, in Panel B, the "T off" was measured prior to the T wave crossing the dotted line. It is unclear how the computer algorithm measured the T waves and how was the algorithm validated. Accurate measurement of the QT interval is important to this and all QT interval studies. Unfortunately, it is notoriously difficult to determine the end of T wave. Serious investigators on QT interval measurements still rely on manual analyses, and the exact method of QT interval determination is extensively explained and documented in the manuscript.

The authors used their data to support the “inherent susceptibility to dysrhythmia”, yet the only arrhythmia was that induced by rapid electrical stimulation in ex-vivo preparation without autonomic innervation. Induction of arrhythmia by electrical stimulation does not require the assistance of autonomic nervous system. It cannot provide strong support to the conclusions.

Reviewer #3 (Remarks to the Author):

The authors studied a very important physiological control mechanism with potential implications for the development of CV diseases, particularly arrhythmias. They show that the cardiomyocyte clock drives rhythms in firing rate and cellular excitability within the SA node and ventricular myocardium, but less so in the AV node, with the latter being more sensitive to the circadian input. Also ventricular repolarization time is rhythmic, but when corrected for HR, ventricular myocardium appears largely driven passively by the rhythm in HR, rather than receiving circadian input from the brain. Thus the contributions of circadian inputs from the autonomic nervous system and the cardiomyocyte clock to the SA and AV nodes differ, and this renders the heart more susceptible to arrhythmias due to an imbalance between circadian input dynamics and abrupt shifts in sleep-wake timing.

Specific comments

The authors demonstrate convincingly in both humans and mice that circadian inputs from the autonomic nervous system and the cardiomyocyte clock regulate the function of SA and AV nodes with many similarities between the two species. Then they exploit a mouse model in which a protein known to contribute to the cardiomyocyte clock was knocked out to show directly the importance of cardiomyocyte clock. I do not have any specific comments to the data as presented. My major concern is that the study is largely descriptive and there is no mechanism provided of how the cardiomyocyte clock specifically contributes to the modulation of cardiac function in the different regions and thus there is no mechanistic insight. Without such information there is no clear conceptual novelty and enthusiasm of this reviewer for this manuscript remains rather limited.

We thank the reviewers for their careful consideration of our manuscript. We are pleased to now submit a revised manuscript which includes exciting new data, additional supporting work, and a revised text. We believe that this has greatly improved the paper and has addressed the comments raised by the reviewers. We provide a detailed point-by-point response to the reviewers' comments below.

Reviewer #1

“Distinct circadian mechanism govern rhythms in cardiac electrophysiology and susceptibility to dysrhythmia” by Hayter and colleagues works to determine how circadian clocks in the heart and brain set the rhythms in the sinoatrial node (SAN) and the atrioventricular node (AVN). They found using a combination of human and mouse studies- circadian inputs to the SAN and AVN are uncoupled during circadian misalignment by behavioral interventions. They use pharmacological approaches to show the different roles of the autonomic nervous system and intrinsic cardiac clock have on day-night rhythms in PRseg and RR intervals. Importantly, disruption of the circadian clock in the heart appears to decrease susceptibility to tachycardia in mice. Conceptually this is an interesting study. The strengths are the combined approaches to show relevance to humans and possible mechanisms in mice.

We thank the reviewer for their support of the manuscript.

1. The authors suggest they are assessing SAN activity by measuring the RR interval (the interval between ventricular depolarizations). Can the authors include the PP interval in their analysis? How does the PP interval change in the longitudinal human and mouse ECG studies? Is it the same as the RR interval?

We appreciate the reviewer's comments related to using PP and/or RR intervals as representing SAN activity. We have now run these analyses across our different data-sets. Through both human and mouse studies, PP interval (from onset of the P wave to the onset of the next P wave) mirrors the daily profiles and responses observed for RR interval. This includes displaying a clear PP-interval diurnal rhythmicity, which is strongly influenced by sleep deprivation and subsequent recovery nap in humans (**Response Figure 1A**). In mice, we again see robust diurnal rhythmicity in PP interval virtually identical to that of RR interval (as evident when PP is subtracted from RR) (**Response Figure 1B-D**). Importantly, no time of day influence over RR/PP difference is observed; again, confirming that measures of PP and RR intervals provide an equivalent circadian read-out of sinus node activity. Finally, we observed a similar pattern of response in PP interval during total autonomic blockade. Specifically, in control mice, a strong time of day rhythm is observed in PP interval under complete autonomic blockade, but this is lost in mice lacking *Bmal1* (and therefore clock activity) in cardiomyocytes (**Response Figure 1D**). Since the PP and RR intervals are so closely matched in human and mouse ECGs, we have not found it necessary to add PP interval plots to the main manuscript.

Response figure 1. PP interval behaves similarly to RR interval in humans and mice.

A. PP interval Z scores across baseline (black) and total sleep deprivation (TSD, orange) days, and quantification across mid-night and nap windows (right), equivalent to Figure 1.

B. RR interval and PP interval across 24hrs in wild-type mice. Average across 3 days.

C. Difference between RR and PP intervals does not vary across the day in mice.

D. Mirroring RR interval, PP interval shows a time of day difference under complete autonomic blockade in control mice, but not in those lacking *Bmal1* function in cardiomyocytes (equivalent to Figure 4C). Data presented as mean \pm SEM, * $p < 0.05$, *** $p < 0.001$.

2. Figure 3 shows longitudinal changes in ECG recordings. How are the RR and HR data different? Is the HR calculated using RR or is the probe actually detecting HR by another method? Since this study is focused on SAN and AVN what is the purpose of showing the QT interval?

We apologise for any lack of clarity in the original manuscript. Throughout our studies HR has been derived directly from our measured RR intervals. We have now added further detail in the methods section to clarify this point (line 424). We acknowledge that this results in essentially the same data being represented twice. However, given the broad readership of *Nature Communications* and the inherent clarity in reporting *heart rate*, we felt that it was appropriate to report the data in both HR and RR interval.

We agree with the reviewer that ventricular depolarisation/re-polarization time is not the primary focus of our studies. However, we believe that it is important to include QT interval in the work. QT interval is highly dependent on heart rate (as shown here, and by many previous studies). Therefore, it serves as a useful parameter with which to compare how other parameters (most notably PR segment) respond to changes in behavioural state and heart rate. Secondly, QT interval duration is of wide clinical interest and reporting potential contribution of the circadian system to its regulation is important. Indeed, there has been debate within the literature as to whether QT interval exhibits a circadian component (beyond the influence of day/night rhythms in heart rate, *per se*). Our studies suggest that in both mice and humans, any true circadian influence on QT interval is minor when compared to the profound influence of time-of-day differences in heart rate. Finally, our studies also detail the sensitivity of the heart to ventricular tachycardia (VT) across time of day and with genetic targeting of the cardiomyocyte clock. Given that prolonged QT interval is associated with increased prevalence of VT in humans, it is also important to show that none of our perturbations profoundly lengthen QT interval in the mice or humans. For these reasons, we have kept all of the results related to QT interval in the manuscript. We acknowledge that inclusion of QT interval does add complexity to the analyses. This is addressed in more detail below, in response to Reviewer 2's comments.

3. Figure 3A. Does the daily change body temperature better correlate with changes in activity similar to RR or is it similar to the PRseg? Based on Figure 3F and 3G body temperature seems to change to the new light cycle similar to PRseg. – is this true? Can you show this more directly?

The reviewer raises an interesting observation, and we apologise that this was not more directly addressed in the original manuscript. Indeed, the reviewer is correct in their observations; rhythms in body temperature do parallel the rhythmic profile of the PR segment. Both parameters show a slow re-entrainment to the new light cycle. We have now included representative profiles and group mean analyses of body temperature during the acute 9hr advancing shift in the LD cycle (included in revised **Supplemental Figure 4**). Please note that the body temperature rhythm has been inverted in Panel A to facilitate comparison with RR interval and PR segment.

Body temperature rhythms are closely linked to central clock activity, and the slow adaption to the 9hr advancing shift reflect the gradual re-entrainment of the central clock to changes in the ambient light-dark cycle. That PR segment is found to have a similar temporal response to that of body temperature reinforces our conclusions that rhythms in AV nodal activity are imposed by a central clock via time-of-day differences in autonomic tone. This is highlighted within the results section text (lines 181-184) of the revised manuscript.

4. Figure 5B-5E are confusing. These are neonatal cardiomyocytes. Are the authors suggesting the intrinsic electrical activity in the cells is analogous to the node? Please clarify or justify this assertion. Have the authors tried a similar study using the alpha myosin heavy chain *Bmal1^{fl/fl}* mouse neonatal cardiomyocytes?

We apologise for a lack of clarity in the original manuscript. We fully agree with the reviewer that neonatal cardiomyocytes are not analogous to pacemaking cells of the SA node. As discussed in more detail below (response to reviewer 3), we now include additional studies specifically addressing circadian mechanisms

within the sinus node and ventricular myocardium (now included in revised **Supplemental Figures 5 and 7**). Specifically, we now show that a number of influential ion channels (which are known to contribute to SA node pacing) exhibit circadian rhythms in expression within the sinus node (**Supplemental Figure 5**). Gene rhythms tend to peak around the transition to the animals' active phase (ZT12), consistent with rhythms in heart rate and decreased RR interval recorded at ZT12 under complete autonomic blockade. These new studies provide important mechanistic insight into how the circadian clock within the SA node shapes nodal electrical activity across the day and night.

Nevertheless, we believe that our studies of neonatal cardiomyocytes also provide useful insight into the fundamental role of the circadian clock in setting intrinsic rhythms in excitability in cardiomyocytes. The accompanying manuscript from Dr John O'Neill's laboratory (LMB Cambridge) explores the fundamental biochemical mechanisms which underlie these clock-dependent rhythms in excitability. This work demonstrates the clock driven daily rhythms in protein synthesis, mTORc signalling and compensatory changes in ion transport lead time-of-day dependent fluctuation in membrane potential and spontaneous firing. We have clarified the relevance of these studies more directly in the manuscript text (line 223). We have not undertaken these studies in the *Bmal1* knockout cells. Although potentially informative, we do not believe that these studies would impact the overall conclusions of the manuscript. Given the circumstances in the UK over the past months (which included restrictions on breeding and experimentation across our labs at the University of Manchester and the Laboratory of Molecular Biology Cambridge), we did not feel that these studies were essential.

5. Figure 5F – is this representative? VT is defined as 4 consecutive premature beats following stimulation. How long did the VT normally last after stimulation in the hearts? Which animal was this from?

The figure originally used in Figure 5F was an example of sustained tachycardia, which we hoped would illustrate the difference between normal pacing and the aberrant firing observed during VT. However, in reviewing all of the pacing responses, we acknowledge that this example is not representative of typical responses observed across the study. We apologise for the lack of clarity and have now replaced this recording with a more representative trace (taken from a control mouse at ZT12). Arrhythmia scoring was carried out in accordance with the Lambeth convention, and responses to pacing which resulted in ≥ 4 premature ventricular complexes (PVC) were scored as an episode of VT. The vast majority of VT responses across all groups showed between 4-10 PVCs (~85%). We have included additional representative traces for the reviewers' benefit (**Response Figure 2**).

Response Figure 2. Individual Langendorff traces. Traces have been aligned to the final electrical stimulus and show VT before returning to sinus rhythm. Arrows highlight premature ventricular complexes.

6. Are there any obvious structural differences in the hearts (size or fibrosis) of *Bmal1^{f/f}* and alpha myosin heavy chain *Bmal1^{f/f}*?

We did not observe any overt structural differences in the hearts between control and cardiomyocyte *Bmal1* knockout mice, nor did we find any decrease in survival in this line. However, previous studies have reported an age-related increase in cardiac fibrosis and hypertrophy in these mice, which becomes evident at 28 weeks of age (Young et al 2014 J Biol Rhythm 29(4):257-76. doi: 10.1177/0748730414543141; Ingle et al 2015 Am J Physiol. Heart & Circ Phys. 2015. 309(11):309: H1827–36 doi.org/10.1152/ajpheart.00608.2015). Ingle et al 2015 reported a minor increase in collagen expression at 8 weeks of age, but this did not translate to any functional changes within their studies. Moreover, the same group (Young et al 2014) reported there were no signs of fibrosis in 12 week old animals, and neither study found evidence of structural changes in younger animals. Finally, Young et al found little evidence of ventricular wall thinning or diastolic diameter, even in animals 36 weeks old. We have restricted our experimental studies to mice aged between 10-16 weeks; thus well before the onset of any previously reported pathology. This is confirmed by the fact that we do not find evidence of altered expression of genes related fibrosis and hypertrophy, or any differences in heart weights (**Response Figure 3**). We do not believe that late onset pathology in the mice have impacted our studies.

Response Figure 3. Gene expression in cardiac tissue from *Bmal1^{f/f}* and *aMHC^{cre}Bmal1^{f/f}* animals shows no evidence of hypertrophy or fibrosis. A. Relative gene expression in ventricular tissue collected from male mice at 12 weeks of age (n=6/genotype). None of the genes were found to be significantly different (adj P>0.1). **B.** Heart weights from 10 (black) and 17 (orange) week old littermate (n=3/age) control and cardiomyocyte-specific *Bmal1* knockout animals. (p>0.1)

Reviewer #2

The authors performed studies in humans and mice. The autonomic nervous system was perturbed using sleep deprivation in humans and drugs (metoprolol and atropine) in mice. The authors concluded that “Our findings reveal a functional segregation of circadian control across the heart’s conduction system and inherent susceptibility to dysrhythmia.” There are several major methodological issues related to this work. The authors measured a single lead ECG. The P, R and QT intervals were automatically detected by a computer algorithm. There are limitations.

We appreciate the consideration and comments made by the reviewer with regards to our ECG analyses and automated analyse approaches. Below we address the comments related to our interpretation and analyses of specific ECG features. Before doing so, we take this opportunity to explain our automated analysis approach more generally. Firstly, we have included with this resubmission the Matlab based programme used for the automated analyses and example ECG recordings from our studies, for the benefit of the reviewers. This would also be uploaded onto GitHub upon acceptance of the manuscript, thereby providing unlimited public access.

As we detail in the methods section, our automated analysis has a number of advantages with respect to deriving accurate and robust ECG feature identification and measurement over longitudinal datasets. The ECG waveform analyses first incorporates a number of quality control features (e.g. baseline stability, signal to noise variation, etc), as well as beat template matching to filter the recordings. Template matching involving a beat-to-beat averaging process (across 5 minute time bins), which defines a robust waveform to which all individual beats within the bin are matched and discriminated. This provides both additional quality control for exclusion of noise, as well as ECG feature measures across both individual beats and 5 min bin-derived averages which dramatically increases signal to noise ratio. As reported in the manuscript, the automated analyses have been thoroughly validated by manual scoring of random sections of ECG recording.

Nevertheless, we understand the reviewer’s caution with regard to automated analyses of ECG recording. Therefore, we have sought additional expertise in cardiac electrophysiology and clinical ECG analyses through Dr Luigi Venetucci (Consultant Cardiologist, Central Manchester University Hospitals NHS Foundation Trust, Manchester, UK). Dr Venetucci was provided with unlabelled extracts of ECG recording from our human sleep study, which he subsequently identified and measured ECG features. We then compared these measures against the automated analyses. As shown in **Response Figure 4**, the expert manual determination was very similar to the automated analyses.

Response Figure 4. Automated ECG measurements do not differ significantly from expert manual measurements. PR, QRS and QT intervals derived from a sweep from each control group individual from Study 1 compared to manual measurements from the average sweep waveform (Adj $p > 0.1$).

1. First, the PR interval is generally measured from the onset of P wave to the onset of QRS complex. It is used to represent the time needed to propagate the impulse from sinus node through AV node and reach the ventricle. However, the PR segment used by the authors measured from the end of the P wave to sometime after the onset QRS complex, when ventricular depolarization has already occurred. The end of P wave is dependent on the completion of the atrial depolarization. Generally the left atria and/or pulmonary veins are depolarized last. The end of P wave is not determined by the AV nodal conduction. Rather, it depends on the impulse propagation within the atria.

We acknowledge that the features used by our analyses to define ‘AV nodal conduction delay’ are less accurate (in terms of isolating AV node conduction, *per se*) than could be achieved with more sophisticated or invasive measures (e.g. His bundle electrography). Using the offset of the P wave does underestimate the AV nodal delay, but it does limit the impact of atrial depolarisation (which would occur if we used onset of the P wave; i.e. PR interval). This would include the time taken for the impulse to travel from the SA node, through the atria to arrive at the AV node. Indeed, we observe clear changes in P wave duration in response to sleep deprivation and mistimed sleep (Fig S2C), which clearly influences measures of PR interval. We therefore opted for the P wave offset to minimise this effect, while making a conservative estimation of AV nodal delay. This distinction is less influential in our mouse studies as the mouse P wave constitutes a significantly smaller proportion of the PR interval than in humans. Finally, as our studies all involve within-individual comparisons (i.e. comparing longitudinal data from the same participant/animal), the absolute quantification of ECG feature duration becomes less critical in comparison to relative changes observed within the participants.

2. The end point of the PR segment was measured at the negative deflection of the QRS complex, when the events in the AV node has already ended. It is not a good estimation of the time when the impulse first crossed the AV node.

Here, we had originally used the negative deflection of the Q wave for robustness of feature discrimination. Given the reviewer’s concerns, we have adjusted the automated detection algorithm to provide a much more accurate identification of Q onset in the human ECG recordings. We have re-run our analyses and updated all relevant figures; no individual findings or overall conclusions have been affected by the change. Within the mouse studies, the differences measured between manually measured Q onset and peak

negative deflection at Q are extremely small and do not change across the day/night. We have therefore not altered these analyses.

3. *Finally, the onset of QRS complex cannot be determined by a single lead ECG. Depending on the electrical axis, some ECG leads would record earlier onset of the QRS complex than others. That earliest onset should be used to close the PR segment.*

We agree that Q onset can vary slightly dependent on electrical axis. Eisenburger et al. (Europace 2010, 12, 119-123) showed that in the absence of pre-excitation the average PR dispersion on a 12 lead ECG is 1 to 5ms, which is approximately 5% of PR segment. However, we emphasize that we are comparing longitudinal data from individuals with consistent electrode placement throughout the study and mice with electrodes fixed internally; therefore, fine differences in absolute PR segment measures are less impactful to the overall conclusions of the study.

4. *The authors did not explain how the QT interval was measured. The “T off” in supplemental figure 1, Panel A, shows it is the time when T wave crosses the dotted line. However, in Panel B, the “T off” was measured prior to the T wave crossing the dotted line. It is unclear how the computer algorithm measured the T waves and how was the algorithm validated. Accurate measurement of the QT interval is important to this and all QT interval studies. Unfortunately, it is notoriously difficult to determine the end of T wave. Serious investigators on QT interval measurements still rely on manual analyses, and the exact method of QT interval determination is extensively explained and documented in the manuscript.*

We apologise for any lack of detail or clarity with respect to the QT measurements. We have now expanded the methods section describing ECG measures (lines 414-422). In our original analysis, T wave offset was calculated as the point at which the ECG trace crossed the isoelectric line. While this measure can underestimate QT duration in our human data, it is incredibly robust to EMG noise. Nevertheless, based on the reviewer's comments we have made a fine adjustment to the method that the analysis algorithm used to define T offset. Specifically, we now use the first positive deflection to the right of the T wave crossing the isoelectric line where appropriate. This new analysis shows very close correlation to manually defined and measured QT intervals (**Response Figure 4**, above). As a result we have reanalysed all of the human ECG recordings and updated all of the respective figures. There has been no change to any of the findings of the paper.

5. *The authors used their data to support the “inherent susceptibility to dysrhythmia”, yet the only arrhythmia was that induced by rapid electrical stimulation in ex-vivo preparation without autonomic innervation. Induction of arrhythmia by electrical stimulation does not require the assistance of autonomic nervous system. It cannot provide strong support to the conclusions.*

The reviewer was right to make this point, and we thank them for pushing us to go further with our studies. We have now undertaken an *in vivo* assessment of susceptibility to bidirectional ventricular tachycardia in control and cardiomyocyte specific knockout mice. Using an established model in which bidirectional VT can be induced in susceptible mice using caffeine and adrenaline {Cerrone et al 2005, 2007 Circ Res}, we now report a strong time of day difference in the induction of bidirectional VT in otherwise healthy mice and a relative protection from arrhythmia in mice lacking Bmal1 in cardiomyocytes (**Figure 6** of the revised manuscript). These findings parallel our ex vivo Langendorff findings, and reveal a critical advance in our understanding of how the inherent susceptibility of the heart to arrhythmia changes across the day and night.

Reviewer #3

The authors studied a very important physiological control mechanism with potential implications for the development of CV diseases, particularly arrhythmias. They show that the cardiomyocyte clock drives rhythms in firing rate and

cellular excitability within the SA node and ventricular myocardium, but less so in the AV node, with the latter being more sensitive to the circadian input. Also ventricular repolarization time is rhythmic, but when corrected for HR, ventricular myocardium appears largely driven passively by the rhythm in HR, rather than receiving circadian input from the brain. Thus the contributions of circadian inputs from the autonomic nervous system and the cardiomyocyte clock to the SA and AV nodes differ, and this renders the heart more susceptible to arrhythmias due to an imbalance between circadian input dynamics and abrupt shifts in sleep-wake timing. The authors demonstrate convincingly in both humans and mice that circadian inputs from the autonomic nervous system and the cardiomyocyte clock regulate the function of SA and AV nodes with many similarities between the two species. Then they exploit a mouse model in which a protein known to contribute to the cardiomyocyte clock was knocked out to show directly the importance of cardiomyocyte clock. I do not have any specific comments to the data as presented.

We would like to thank the reviewer for their support of the importance and robustness of our studies.

My major concern is that the study is largely descriptive and there is no mechanism provided of how the cardiomyocyte clock specifically contributes to the modulation of cardiac function in the different regions and thus there is no mechanistic insight. Without such information there is no clear conceptual novelty and enthusiasm of this reviewer for this manuscript remains rather limited.

We understand the reviewers request for additional mechanism. In our original manuscript, we had already detailed how segregation of circadian influence over SA and AV nodal function involves a differential influence of local circadian clock function (more dominant at the SA node) and rhythmic autonomic input (highly dominant over AV node function). This was demonstrated with in vivo ECG analyses, autonomic manipulations and ex vivo cardiac electrophysiology. In light of the reviewer's comment, we have investigated further how circadian rhythms in electrical activity may be delivered at the SA node. Through microdissection of nodal tissue across the day/night cycle, we now reveal for the first time a pronounced temporal oscillation of ion channels critical to pacesetting at this site (**Supplementary Figure 5** of the revised manuscript). This is in keeping with previous studies using whole heart or ventricular tissue, which demonstrate circadian rhythms in ion channel expression {Schroder et al 2015, Heart Rhythm, 12(6):1306-14; Tong et al 2013 Biol Rhythm Res, 44(4):519-30; Schroder et al 2013, Am J Physiol Cell Physiol, 304(10):C954-65; Jeyaraj et al 2012, Nature, 483(7387):96-9; Yamashita et al 2003, Circulation, 107:1917-22}.

Another important finding in our manuscript is the time of day variation in susceptibility of the heart to arrhythmia, and the apparent protection provided by the loss of Bmal1 in cardiomyocytes. As discussed above this has been demonstrated both by electrical pacing induced arrhythmia *ex vivo*, and in response to catecholamine challenge *in vivo*. To provide insights to mechanisms in the heart which underlie this variation in susceptibility, we have examined ventricular gene expression at the two times of day tested in our arrhythmia studies (ZT0 and ZT12, corresponding to the start of the rest phase and start of the active phase for the mice, respectively). These analyses reveal both time of day (e.g. Kcnh2, Scn5a, cacna1c, adrb1) and circadian clock dependent (e.g. casq1, casq2, cacna1c, kchip2) processes linked to cardiac arrhythmias in humans and mouse transgenic models (**Supplementary Figure 7** in the revised manuscript). The pronounced alteration in calcium handling and overexpression of casq2 are likely to underpin the relative protection from induced arrhythmia observed in the cardiomyocyte specific Bmal1 knockout mouse.

We believe that these new additions provide substantial insight into circadian mechanisms within the heart and have greatly strengthened the manuscript. The accompanying manuscript submitted by Dr John O'Neill (and again included in our resubmission) explores fundamental aspects of rhythmic influences contributing to cardiomyocyte excitability. Their studies complement our work, and together these studies reveal a new understanding of how circadian processes exert a profound influence on cardiac physiology.

REVIEWERS' COMMENTS

Reviewer #1 (Remarks to the Author):

Please provide clear definitions and consistent use for the terms: synchrony, alignment, desynchrony/desynchronization, misalignment, and mistimed. When possible, clarify if this is relative to the internal, behavioral or environmental cycles.

Figure 6S. These are double plots of the same data (as indicated in legend). The units on the X-axis indicating Zeitgeber time needs correction.

Figure 7S. Did you survey genes in the circadian clock mechanism (e.g., Clock, Period, Cry genes)?

Please include a brief Discussion your results as it relates to the findings published by D'Souza et al., A circadian clock in the sinus node mediates day-night rhythms in Hcn4 and heart rate. Heart Rhythm. November 2020.

Reviewer #3 (Remarks to the Author):

The authors satisfactory addressed my comments. I do not have further concerns. Congratulations to this interesting work.

We thank the reviewers for their positive assessment of our manuscript. We are pleased to now submit a revised manuscript, figures and supporting data which addresses the final few comments raised by reviewer 1 and by the editorial staff at Nature Communications. We provide a detailed point-by-point response to the reviewers' comments below.

Reviewer #1

Please provide clear definitions and consistent use for the terms: synchrony, alignment, desynchrony/desynchronization, misalignment, and mistimed. When possible, clarify if this is relative to the internal, behavioral or environmental cycles.

We acknowledge that these terms are used widely in circadian biology, but often without being properly defined. The manuscript text has been edited to remove confusion between these terms:

- We have now removed any use of “synchrony”, “desynchrony” and “desynchronization”.
- We more clearly define the term ‘circadian misalignment’ in the revised manuscript (ln 73-76). This term is widely used where behavioural or environmental rhythms do not match with internal clock timing. For example, during an abrupt change in behavioural routine (as in the simulated shift work protocol in our second human study, or 9hr advancing phase shift study in mice) the internal circadian clock takes time to re-align to the new routine. This creates a temporal misalignment between internal and external/behavioural rhythms.
- Within the manuscript the term ‘mistimed’ is used to describe the deviation from normal sleep and behavioural routine imposed during the human studies (ln 99, 185, 342). We feel that this is clear in context and meaning.

Figure 6S. These are double plots of the same data (as indicated in legend). The units on the X-axis indicating Zeitgeber time needs correction.

We have now corrected the x-axis labelling as the reviewer has suggested.

Figure 7S. Did you survey genes in the circadian clock mechanism (e.g., Clock, Period, Cry genes)?

As shown in Response Figure 1, we have now assessed clock gene expression in the ventricular tissues collected from *Bmal1^{fl/fl}* and *aMHC^{cre}Bmal1^{fl/fl}* mice. All of the clock genes show a significant damping of rhythmic expression (across the two time-points examined) in the targeted mice. Remaining temporal rhythmicity in these tissues was expected due to the selectivity of our genetic targeting to cardiomyocytes. A substantial population of myofibroblasts (and other minor populations) would retain expression of *bmal1* and presumably wider clock function. To highlight the specificity of the genetic targeting, we have also bred the *aMHC^{cre}* mouse to our CRE-dependent clock reporter mouse (*ReverbaStopLUC*), which couples *Reverba* gene expression to luciferase activity in a cell-specific manner. Robust and rhythmic bioluminescence is observed only in the hearts of these mice, with peak (ZT8) and trough (ZT20) luminescence matching the known phase of *Reverba* (Image recorded using IVIS In Vivo Imaging System).

Please include a brief Discussion your results as it relates to the findings published by D'Souza et al., A circadian clock in the sinus node mediates day-night rhythms in Hcn4 and heart rate. Heart Rhythm. November 2020.

We were pleased to see this work published as it is complementary to our own studies. Both studies (D'Souza et al and our own) demonstrate a significant contribution of clock function in the sinus node to daily rhythms in inherent heart rate. We have highlighted this paper in our revised manuscript (ln 219-220) as suggested.

Response Figure1. Cardiomyocyte-selective targeting. **A.** Clock gene expression in ventricular tissue collected from *bmal1^{fl/fl}* and *aMHC^{cre}bmal1^{fl/fl}* mice (n=6/group). **B.** Bioluminescence image recorded from conditional *aMHC^{CRE}ReverbaStopLuc* transcriptional reporter mouse line at peak (ZT8) and trough (ZT20) of *reverba* expression.